# Efficient Adaptive Testing via Gradient Path Matching Subset Selection for AI Education

Yan Zhuang [1]  Junhao Yu [2]  Bohou Zhang [2]  Zachary A. Pardos [3]  Jinze Wu [4]  Daoqiang Zhang [1]

## Abstract

Adaptive testing is widely adopted in AI-driven educational assessment systems (e.g., GRE), where the goal is to select an optimal subset of questions from a large question pool to accurately estimate an examinee's ability. A fundamental challenge is that: optimal question subsets are inherently personalized, and solving for them is NP-hard. Recently, it has been framed as a gradient matching problem: aligning gradients between selected subsets and the full question set across the entire ability parameter space. However, such global alignment on entire space is computationally expensive and difficult to scale. In this work, we propose GPM (Gradient Path Matching), a novel framework that instead aligns gradients along possible optimization paths toward the final estimate. By leveraging intermediate gradients as supervision, GPM learns an explicit and generalizable selection algorithm from large-scale data. We provide theoretical analysis on its convergence and scalability. Experiments on both real-world and synthetic datasets demonstrate that it achieves the same estimation accuracy using, on average, 9% fewer questions.

## 1. Introduction

Adaptive Testing is an efficient approach in AI Education for estimating an examinee's ability, aiming to achieve accurate estimation using as few questions as possible. This method has been widely adopted across various human testing systems, e.g., GRE, TOEFL, Duolingo (Van der Linden & Pashley, 2000; Liu et al., 2024a; Yu et al., 2024b).

[1]College of Artificial Intelligence, Nanjing University of Aeronautics and Astronautics, China [2]State Key Laboratory of Cognitive Intelligence, University of Science and Technology of China, China [3]University of California, Berkeley, USA [4]iFLYTEK Co., Ltd, China. Correspondence to: Yan Zhuang <yanzhuang@nuaa.edu.cn>.

*Proceedings of the 43rd International Conference on Machine Learning*, Seoul, South Korea. PMLR 306, 2026. Copyright 2026 by the author(s).

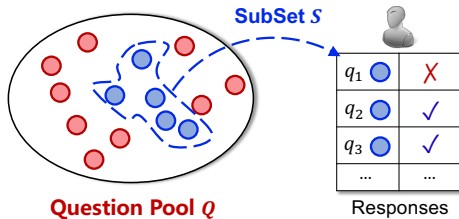

*Figure 1.* An illustration of adaptive testing: selecting an optimal question subset from a large pool for each examinee.

Compared with the time-consuming and burdensome paper-and-pencil tests, adaptive testing has been shown to achieve equivalent measurement accuracy with significantly fewer questions (Lan et al., 2014; Vie et al., 2017). As the scale and availability of expansive question data continue to grow, the demand for efficient and scalable testing systems has become more urgent than ever.

Adaptive testing aims to estimate each examinee's true ability $\theta^*$ accurately by requiring them to answer only a small number of questions. It essentially addresses a challenge of *accuracy and efficiency*. Formally, as shown in Figure 1 adaptive testing is a *subset selection problem*: selecting a small question subset $S \subseteq Q$ from question pool $Q$ to minimize the estimation error:

$$\min_{S \subseteq Q} \|\theta_S - \theta^*\|, \tag{1}$$

where $\theta_S$ denotes the ability estimate computed from the examinee's responses to the selected subset $S$, and $\theta^*$ denotes the (unobserved) true ability. Since $\theta^*$ is not directly observable in practice, it is commonly approximated by the ability estimate obtained using the full question set, i.e., $\theta^* \approx \theta_Q$. In principle, $\theta_Q$ is the closest available proxy to $\theta^*$, and is therefore used as the target signal when optimizing subset selection strategies (Liu et al., 2024a; Zhuang et al., 2025).

However, the above problem is fundamentally unsolvable. Each examinee has a different true ability and provides different responses, which implies that the optimal subset $S$ is also individual-specific. Consequently, adaptive testing involves solving unlimited *personalized NP-hard* subset selection problems, one for each examinee. Numerous approaches have been attempted: from heuristic strategies that select questions based on informativeness / represen-

tativeness / diversity (Lord, 2012; Bi et al., 2020), to more recent reinforcement learning methods (Wang et al., 2023a; Ghosh & Lan, 2021; Ma et al., 2023). To explicitly approximate the optimal subset, this problem has been successfully framed as a gradient matching task: minimizing the gradient discrepancy w.r.t. the parameter $\theta$ between the subset $S$ and the full set $Q$ across the entire ability space (Zhuang et al., 2023). This ensures that the subset ultimately yields the same ability estimate. While intuitive, it requires approximating gradients over a continuous space, which is computationally expensive and demands extensive approximations.

In contrast to prior approaches that aim to match gradients across the entire ability space, we introduce a novel perspective: instead of global alignment, *it is sufficient (and often more effective) to ensure alignment along the gradient descent path toward the final parameter estimate*. Building on this insight, we propose GPM (Gradient Path Matching), which learns a small subset or question selection algorithm from large-scale data. GPM leverages intermediate gradient information along the optimization path as supervision, enabling the training of a generalized and explicit selection algorithms. We further provide theoretical analysis of GPM's convergence and scalability, demonstrating its advantages in both efficiency and practical deployment.

To validate its effectiveness, we conduct experiments on three real-world educational datasets from different testing platforms. Empirical results show that this framework achieves SOTA performance compared with other implicit methods. It consistently outperforms existing methods by reducing test length (subset size), requiring 9% fewer questions to achieve the same estimation accuracy. Beyond efficiency, it also shows clear advantages in terms of convergence speed, robustness (especially under data sparsity and high-noise conditions), and interpretability.

## 2. Related Works on Adaptive Testing

Adaptive testing has been widely used in human assessments, particularly for evaluating abilities and subject-specific knowledge. A practical testing system typically consists of two key components: **1)** a user model, often based on the famous psychometric model Item Response Theory (IRT) (Lord, 2012), which estimates the examinee's ability ($\theta$) from their responses on the selected subset; and **2)** a selection algorithm selects a small subset of questions from the entire question pool based on specific criteria, such as informativeness or representativeness.

**(1) Item Response Theory.** IRT is the most widely used framework in psychometrics for modeling examinee ability and predicting response outcomes (Van der Linden & Pashley, 2000). It estimates an exami-

nee's latent ability based on their response data, typically via gradient-based optimization. A commonly used implementation is the two-parameter logistic model: $p(\text{correct response to question } i) = \sigma(\alpha_i(\theta - \beta_i))$, where $\alpha_i$ and $\beta_i$ are the discrimination and difficulty parameters of question $i$, and $\theta$ is the examinee's ability. These item parameters are typically pre-calibrated (Lord, 2012). Recent work has extended IRT by integrating it with neural networks and cognitive diagnosis models to capture more complex examinee-question interactions (Wang et al., 2022; Yu et al., 2024b; Wang et al., 2023b).

**(2) Selection Algorithms.** The selection algorithm determines which questions to present, aiming to estimate ability accurately with as few questions as possible. Existing algorithms generally fall into two major categories: The first category relies on *implicit* algorithms, e.g., heuristic strategies based on informativeness, representativeness, or diversity (Lord, 2012; Bi et al., 2020; Chang & Ying, 1996; Rudner, 2002). More recently, genetic algorithm (Li et al., 2025), reinforcement learning (Zhuang et al., 2022; Wang et al., 2023a) and meta-learning techniques (Ghosh & Lan, 2021; Yu et al., 2024a) have been applied to train selection strategies on large-scale response data. These data-driven methods aim to minimize prediction loss on held-out response data, indirectly minimizing the ability estimation error; The second category relies on *explicit* algorithms that directly target the adaptive testing objective. These methods formulate question selection as a subset selection problem (Zhuang et al., 2023; Liu et al., 2024b), aiming to minimize the gradient discrepancy between the selected subset and the question pool. Compared to implicit methods, the explicit offer theoretical guarantees that the estimate can efficiently approach $\theta^*$. Their reliability and validity make them particularly suitable for high-stakes testing systems. The proposed GPM in this paper follows the explicit optimization paradigm through gradient path matching.

## 3. Problem Definitions of Adaptive Testing

Adaptive Testing aims to accurately estimate each examinee's true ability $\theta^*$, by selecting a small subset of questions $S \subseteq Q$ from a larger question pool $Q$.

**Preliminaries.** The size of the selected question set $S$ determines the test length $T$, i.e., $T = |S|$. The examinee answers these selected questions, producing response data $\tau_T = \{(q_1, y_1), ..., (q_T, y_T)\}$, where $S = \{q_i\}_{i=1}^{T} \subseteq Q$ is the question set selected by the algorithm, and $y_i \in \{0, 1\}$ represents response label (correct/incorrect). Then, the examinee's ability can be estimated by minimizing the empirical risk (e.g., cross-entropy) over the ability space $\Theta$:

$$\theta_S = \arg\min_{\theta \in \Theta} \sum_{i \in S} \ell(q_i, y_i; \theta), \qquad (2)$$

where $\ell(q_i, y_i; \theta) = -y_i \log p_\theta^i - (1 - y_i) \log(1 - p_\theta^i)$ denotes the loss of the observed response $(q_i, y_i)$ given an examinee with ability $\theta$. $p_\theta^i \in (0, 1)$ predicts the probability of a correct response to $q_i$, and the specific form of $p_\theta$ is determined by the IRT model. Formally, from a global perspective, the goal of adaptive testing is defined as follows:

**Definition 3.1** (Definition of Adaptive Testing). Selecting a subset $S \subseteq Q$ such that the ability estimate $\theta_S = \arg\min_\theta \sum_{i \in S} \ell(q_i, y_i; \theta)$ approximates the true ability $\theta^*$:

$$\min_{S \subseteq Q, |S| = T} \|\theta_S - \theta^*\|. \tag{3}$$

Unfortunately, directly solving the above optimization problem is *infeasible* due to: **(1)** It involves a complex nested optimization structure. For each candidate $S$, one must solve an inner optimization problem, which requires repeatedly computing $\theta_S$ (through massive gradient descent based on $\sum_{i \in S} \nabla_\theta \ell(q_i, y_i; \theta)$); **(2)** The optimal subset $S$ varies across examinees, turning the subset selection into a personalized combinatorial optimization problem. When scaled to a large number of examinees, this results in solving unlimited personalized NP-hard problems.

These challenges make existing selection algorithms incapable of directly solving this subset selection problem. To explicitly address this and ensure the reliability of ability estimation, (Zhuang et al., 2023) reformulates the subset selection task as a gradient approximation problem. Building on it, we further establish the following theoretical connection between gradient approximation and estimation error:

**Lemma 3.2** (Gradient Approximation Bounds Estimation Error). *Let $\theta_S$ and $\theta^*$ denote the estimated ability parameters based on a subset $S \subseteq Q$ and the full set $Q$, respectively. When using IRT to estimate examinee ability (L2-norm loss), there exists a constant $C > 0$ such that the estimation error is bounded by the maximum gradient approximation error:*

$$\min_S \|\theta_S - \theta^*\| \leq C \cdot \min_S \sup_{\theta \in \Theta} \|\nabla_S(\theta) - \nabla_Q(\theta)\|, \tag{4}$$

*where $\nabla_S(\theta) = \sum_{j \in S} \gamma_j \nabla \ell(q_j, y_j; \theta)$, with $\gamma_j$ being a scaling weight that adjusts each selected question's gradient contribution; and $\nabla_Q(\theta) = \sum_{i \in Q} \nabla \ell(q_i, y_i; \theta)$.*

The proof is provided in Appendix B. Lemma 3.2 transforms the original nested optimization problem into a more tractable one by focusing on minimizing the gradient approximation error w.r.t. $S$. This formulation considers the *worst-case error* over the entire parameter space $\Theta$, i.e., $\sup_{\theta \in \Theta}$. This ensures that the selected subset $S$ provides uniformly good gradient approximations across all possible ability levels. This is intuitive (top of Figure 2): when the gradients of $S$ closely match those of $Q$ at every point in $\Theta$, *gradient descent initialized from any point will be guided in a direction that remains close to the true optimum $\theta^*$.*

Obviously, despite its theoretical appeal, such gradient-matching approach can be computationally prohibitive. Selecting even a single optimal question requires evaluating gradient discrepancies for all $|Q|$ questions across the entire parameter space $\Theta$, resulting in a complexity of $O(|Q||\Theta|)$. Therefore, the practical efficiency of this method is highly sensitive to the dimensionality and continuity of $\Theta$.

## 4. Iterative Gradient Path Matching

In fact, enforcing gradient matching over the entire space $\Theta$ is somewhat overly restrictive. Our idea is very simple: **matching the optimization convergence path that leads to the correct result, rather than exhaustively searching the entire parameter space.** Specifically, in stochastic gradient descent, assuming a sufficient number of iterations (denoted by $K$), the optimal ability value can be reliably obtained: for $k = 0, 1, ..., K - 1$

$$\theta_Q^{(k+1)} = \theta_Q^{(k)} - \eta_\theta \nabla_Q(\theta_Q^{(k)}), \tag{5}$$

where $\eta_\theta$ is the learning rate, and the iterative process aims to converge to an approximately optimal ability estimate, denoted as $\theta_Q^{(K)} \approx \theta^*$. A similar gradient update is performed over the selected subset $S$:

$$\theta_S^{(k+1)} = \theta_S^{(k)} - \eta_\theta \nabla_S(\theta_S^{(k)}). \tag{6}$$

This process generates two gradient paths: $\{\nabla_S(\theta_S^{(k)})\}_{k=0}^K$ and $\{\nabla_Q(\theta_Q^{(k)})\}_{k=0}^K$. For simplicity, during the optimization, we keep the ability parameters on $Q$ identical to those on $S$, i.e., $\theta_Q^{(k)} = \theta_S^{(k)}$. Thus, with a fixed initialization of $\theta$, minimizing the discrepancies between the two gradient paths can lead to the same optimization outcome. The objective becomes:

$$\min_S \sum_{k=1}^K \left\| \nabla_S(\theta_S^{(k)}) - \nabla_Q(\theta_S^{(k)}) \right\| := \min_S \sum_{k=1}^K \Delta\big(S, \theta_S^{(k)}\big),$$

where $\Delta(S, \theta_S) = \|\nabla_S(\theta_S) - \nabla_Q(\theta_S)\|$ represents the discrepancies in gradient paths and is a function of both $S$ and $\theta_S$, and $\theta_S$ is also a function of $S$. This formulation aligns with the notion of data efficiency in machine learning (Mirzasoleiman et al., 2020; Lei & Tao, 2023). As illustrated in Figure 2, instead of matching gradients across the entire $\Theta$, this optimization focuses on a *feasible optimization trajectory*, significantly reducing the search space and problem complexity. By leveraging its intermediate gradients, the method enables continuous supervision and adjustment. The following theoretical result further supports the above idea:

**Lemma 4.1.** *Let $\{\theta_S^{(k)}\}_{k=1}^K \subset \Theta$ be a set of ability estimates such that for every $\theta \in \Theta$, there exists some $k \in \{1, \dots, K\}$ with $\|\theta - \theta_S^{(k)}\| \leq \varepsilon$ and $K \geq 1$.*

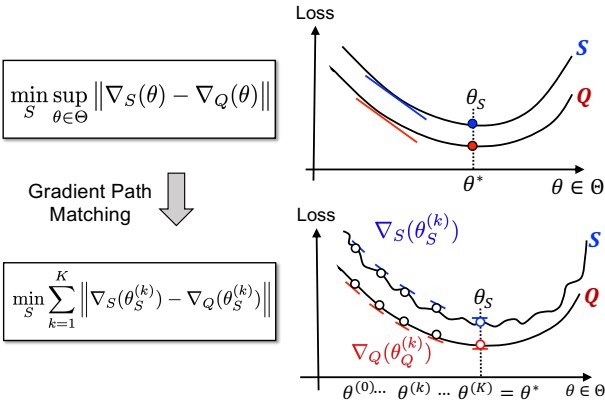

$$\min_S \sup_{\theta \in \Theta} \|\nabla_S(\theta) - \nabla_Q(\theta)\|$$

Gradient Path
Matching

$$\min_S \sum_{k=1}^{K} \left\|\nabla_S(\theta_S^{(k)}) - \nabla_Q(\theta_Q^{(k)})\right\|$$

*Figure 2.* Finding a feasible gradient path that ensures the ability estimates on $Q$ and $S$ have the same gradient at each point $\{\theta^{(0)}, \theta^{(1)}, ..., \theta^{(K)}\}$, rather than over the entire space $\Theta$. Both of them result in the same outcome: $\theta_S = \theta^*$.

*When using IRT for ability estimation, there exists a constant $L$ such that:* $\min_S \sup_{\theta \in \Theta} \|\nabla_S(\theta) - \nabla_Q(\theta)\| \leq \min_S \sum_{k=1}^{K} \|\nabla_S(\theta_S^{(k)}) - \nabla_Q(\theta_S^{(k)})\| + L\varepsilon.$

This lemma provides a theoretical justification for using trajectory-based empirical gradient matching as a surrogate objective for approximating the worst-case gradient error (the original objective in Eq. (4)). *The empirical trajectory serves as a finite approximation of the continuous parameter space $\Theta$. The bound becomes tighter as the sample points $\{\theta_S^{(k)}\}$ provide better coverage of $\Theta$ (i.e., as $\varepsilon$ decreases), which is typically improved by longer or more informative optimization trajectories.*

### 4.1. Subset Optimization

To effectively minimize discrepancies in gradient sequences and optimize $S$, we develop a data-driven selection algorithm. It leverages large-scale examinee response data to learn the patterns by which *different questions contribute to gradient discrepancy approximation for diverse examinees*. Here, $S$ represents a parameterized selection algorithm or a differentiable indicator vector of length $|Q|$: $S = [\gamma_1, \gamma_2, \ldots, \gamma_Q]$. $\gamma_i$ represents the probability of selecting each question in $Q$, also serving as the weight for each item in $S$. Such optimization has the following properties:

**Lemma 4.2** (A Lipschitz Continuous Gradient). *The function of gradient sequence discrepancy $\Delta(S, \theta_S)$ serves as the loss function for optimizing $S$. When using L2-norm IRT for estimating ability $\theta$, $\Delta(S, \theta)$ is Lipschitz continuous, and gradient $\nabla_S \Delta(S, \theta_S)$ is also Lipschitz with constant $L_\Delta$. Then, for any $S, S'$:*

$$\|\nabla_S \Delta(S, \theta_S) - \nabla_S \Delta(S', \theta_{S'})\| \leq L_\Delta \|S - S'\|. \quad (7)$$

The proof is provided in Appendix D. The Lipschitz continuity of the gradient ensures that the gradient does not change

---

**Algorithm 1** Optimization Process of GPM

---

**Require:** Examinees response data $\{\tau\}$, number of outer-loop steps $N$, number of inner-loop steps $K$, learning rates $\eta_\theta$ and $\eta_S$ for updating ability and subset.
1: Initialize the parameterized subset $S^{(0)}$.
2: **for** $n = 1, 2, \ldots, N$ **do**
3:     Sample $\tau \sim \{\tau\}$, where $\tau = \{(q_i, y_i)\}$ is the response sequence over the full pool $Q$
4:     Initialize the ability parameters $\theta^{(0)}$.
5:     **for** $k = 1, 2, \ldots, K$ **do**
6:         Sample a subset $S_\tau \sim S^{(n-1)}$.
7:         Update the ability parameters on $S_\tau$:
8:         $\theta^{(k)} \leftarrow \theta^{(k-1)} - \eta_\theta \nabla_{S_\tau}(\theta^{(k-1)})$
9:         Compute gradient discrepancy:
10:        $\Delta(S, \theta^{(k)}) \leftarrow \|\nabla_S(\theta^{(k)}) - \nabla_Q(\theta^{(k)})\|$
11:        Update subset parameters:
12:        $S^{(n)} \leftarrow S^{(n)} - \eta_S \nabla_S \Delta(S^{(n)}, \theta^{(k)})$
13:     **end for**
14: **end for**
15: **Output:** Optimized subset $S \sim S^{(N)}$

---

too rapidly. This is crucial for gradient-based optimization, as it allows for more predictable and reliable convergence behavior (Chi et al., 2020).

The detailed optimization process is outlined in Algorithm 1. The GPM effectively leverages extensive examinee response data $\{\tau\}$ and $\tau = \{(q_i, y_i)\}_i$, which comprises the responses of examinees of varying abilities to the question pool $Q$. At the outer level, the process involves $N$ iterations of optimization, in which the response data are continuously sampled to train $S$. At the inner level, $K$ iterations of ability estimation are performed, where the gradients on both $Q$ and $S$ are computed. The gradient discrepancies in ability optimization are then used to update $S$. To ensure optimization stability, especially during the initial training phase, cumulative gradients can also be used to update $S$ instead of single-step updates: $S^{(n)} = S^{(n)} - \eta_S \sum_k \nabla_S \Delta(S^{(n)}, \theta^{(k)})$. The final optimized $S^{(N)}$ provides the selection probability for each question in $Q$. Based on the required test length, the top $T$ elements are selected, and softmax normalization is applied to compute the corresponding weights $\{\gamma\}$.

### 4.2. Property Analysis

Similar to other data-driven question selection algorithms (Zhuang et al., 2022; Ghosh & Lan, 2021; Yu et al., 2024a), GPM operates efficiently in practical deployments by fixing its well-trained parameters, eliminating the need for real-time parameter updates during examinee testing. This significantly enhances the selection efficiency. More importantly, GPM framework prioritizes the core objective of adaptive testing, i.e., the accuracy of ability estimation

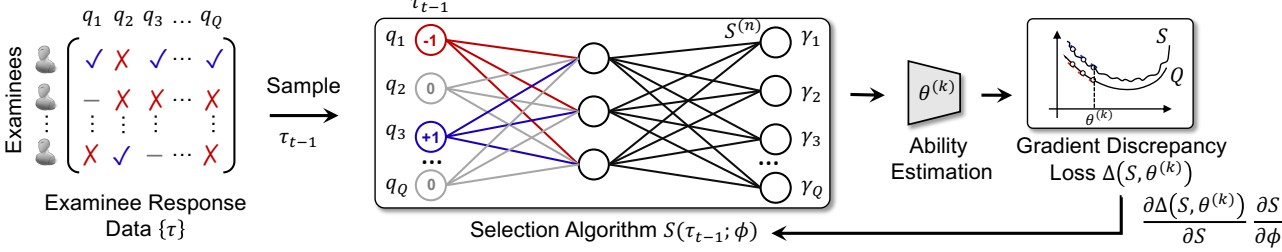

*Figure 3.* Training process of the personalized GPM algorithm. The algorithm dynamically selects the next item based on the examinee's responses at each step. Following the training procedure in Algorithm 1, the previous responses $\{\tau_0, \ldots, \tau_t\}$ are iteratively sampled as input, and the gradients are propagated through $S^{(n)}$ to update the algorithm parameters $\phi$.

instead of score predictions.

**Convergence Performance.** Compared to the previous heuristic algorithm or purely theoretical approximations, this data-driven approach is more aligned with real-world scenarios that utilize multi-step gradient descent for ability estimation, thereby significantly reducing the search space. Moreover, to validate its training efficiency, we further characterize the convergence performance of training GPM:

**Theorem 4.3** (Upper Bound of Gradient Norm). *When optimizing $S$ with a learning rate $\eta_s < \frac{2}{L_\Delta}$ and the ability parameter $\theta$ is optimized to $\theta^{(k)}$, the gradient discrepancy loss $\Delta(S, \theta_S)$ over $N$ steps satisfies:*

$$\frac{1}{N}\sum_{n=1}^{N}\left\|\nabla_S\Delta\left(S^{(n)}, \theta_{S^{(n)}}^{(k)}\right)\right\|^2 \le \frac{2\Delta\left(S^{(0)}, \theta_{S^{(0)}}^{(k)}\right)}{N(2\eta_s - L_\Delta\eta_s^2)}. \quad (8)$$

The detailed proofs can be found in Appendix E. Theorem 4.3 provides an upper bound on the gradient norm during the optimization of the subset $S$. This result guarantees that as the number of iterations $N$ increases, the gradient approaches zero. Thus, the optimization process for $S$ remains stable and controlled. Such convergence property implies that the proposed algorithm will make consistent progress towards minimizing the gradient discrepancy loss.

### 4.3. Implementation of a Personalized Algorithm

The aforementioned algorithm produces a 'global' subset $S$, i.e., a uniform set of questions for all examinees. However, in the practice of adaptive testing, it is crucial to enhance adaptability by personalizing the selection of test questions based on an examinee's performance. To achieve this, a personalized selection algorithm $S(\tau; \phi)$ with parameters $\phi$ is further designed to evaluate the value of remaining questions based on an examinee's previous responses $\tau$.

Specifically, as illustrated in Figure 3, at step $t \in [0, T]$ for a given examinee, their prior $t - 1$ responses (denoted as $\tau_{t-1}$) can be represented as a vector of length $|Q|$, where each element takes values in $\{+1, -1, 0\}$, corresponding respectively to correct, incorrect, and unanswered questions.

The algorithm outputs a probability distribution over all questions in $Q$ (a score vector using a softmax function). Assuming the algorithm $S(\tau; \phi)$ outputs probabilities $S = [\gamma_1, \ldots, \gamma_Q]$, the next question is selected:

$$q_t \sim S(\tau_{t-1}; \phi) = \arg\max_{q \in Q}\left(\frac{\exp(\gamma_q)}{\sum_{q'=1}^{|Q|}\exp(\gamma_{q'})}\right), \quad (9)$$

where $\tau_{t-1} \subseteq \tau$ is serialized from complete $\tau$ to simulate the adaptive testing process in algorithm training. To sample discrete selections while maintaining differentiability and allowing for gradient flow, we employ the common Gumbel-Softmax trick (Jang et al., 2022). After extensive training, all parameters in selection algorithm are fixed during deployment and are not updated online as the test proceeds. Therefore, in practical use the algorithm only needs to compute the score vector $S$ and select the highest-probability question, yielding a per-step complexity of $O(|Q|)$.

The gradient can be updated based on the gradient discrepancy. The gradient from $\Delta$ to $\phi$ can then be computed as:

$$\frac{\partial}{\partial\phi}\mathbb{E}_{\tau\sim\{\tau\}}[\Delta(S, \theta_S)] = \mathbb{E}_{\tau\sim\{\tau\}}\left[\frac{\partial\Delta(S, \theta_S)}{\partial S}\cdot\frac{\partial S}{\partial\phi}\right]. \quad (10)$$

By learning iterative gradient matching through optimization on large-scale data, the algorithm avoids the approximation errors typically introduced by manual design and derivation. The detailed process of paths sampling and the optimization can be found in Appendix A. This question selection algorithm explores various combinations of questions to find the optimal subset based on every context encountered during testing.

## 5. Experiments

In this section, we conduct both quantitative and qualitative experiments on three real-world datasets and synthetic data to evaluate the effectiveness of our proposed GPM.

## 5.1. Evaluation Method

Adaptive testing aims to estimate the examinee's ability accurately with the fewest steps (subset size). There are usually two tasks to verify the performance of different methods;

*1) Simulation of Ability Estimation.* Following standard practice in adaptive testing (Vie et al., 2017), true examinee abilities $\theta^*$ are simulated using responses from the entire dataset, as described by (Bi et al., 2020; Cheng, 2009). These estimated abilities serve as ground truth, and question parameters are pre-estimated from the full dataset and remain fixed. At each simulation step: (1) a selection algorithm picks a question from the pool $Q$; (2) IRT updates the ability estimate based on the simulated response; and (3) estimation accuracy is measured via Mean Squared Error (MSE) between the estimated and true abilities.

*2) Examinee Score Prediction.* To evaluate the quality of ability estimates in predicting examinee responses, the dataset is partitioned into training (70%), validation (20%), and test (10%) sets, ensuring examinee-level splits. Training data initialize IRT parameters and train selection algorithms. For each examinee in validation/testing, responses are split into a candidate question set $Q_i$ and a held-out meta set $M_i$ (Ghosh & Lan, 2021). At each step: (1) a selection algorithm selects a question from $Q_i$; (2) IRT updates the examinee's ability estimate based on the response; and (3) prediction accuracy is evaluated on $M_i$. Results are reported using Prediction Accuracy (ACC) and AUC (Bradley, 1997).

**Datasets.** We conduct experiments on three educational testing benchmark datasets, namely ASSIST, NIPS-EDU, and EXAM. ASSIST (Pardos et al., 2013) is collected from an online educational system ASSISTments and consists of examinees' practice logs on mathematics. NIPS-EDU (Wang et al., 2020) refers to the large-scale dataset in NeurIPS 2020 Education Challenge, which is collected from examinees' answers to questions from Eedi (an educational platform). EXAM is a dataset from iFLYTEK Co., Ltd. that records junior high school students' performances on mathematical exams. The statistics of the datasets are shown in Appendix. The code can be found in: https://github.com/54zy/GPM.

**Experimental Implementation Details.** As 20 is sufficient for the length of a typical test (Liu et al., 2024a), we also fix the max length $|S| = T = 20$. The proposed GPM is a simple two-layer neural network: an observation layer that maps the input $(= |Q|)$ to a 256-dimensional latent space, and an actor layer that maps the latent space to the size of the question pool through a two-layer structure with Tanh. We implement all the methods with PyTorch. We set batch size to 64 and the learning rate to 0.001, and optimize all the parameters using the Adam algorithm (Kingma & Ba,

2015) on a Tesla V100-SXM2-32GB GPU.

## 5.2. Compared Approaches

To verify the generality of GPM, in addition to the traditional **IRT**, we also compare the neural network-based model **NeuralCDM** (Wang et al., 2022). Since our focus is on comparing question selection algorithms, we restrict our experiments to these two representative model families.

For selection algorithm, baselines include the random selection strategy as a simple benchmark; information-based approaches like Fisher Information (**FSI**) (Lord, 2012) and KL-divergence (**KLI**) (Chang & Ying, 1996); uncertainty- and diversity-based Active Learning (**MAAT**) (Bi et al., 2020); meta-learning methods **BOBCAT** (Ghosh & Lan, 2021) and **UATS** (Yu et al., 2024a); reinforcement learning approaches using transformer and graph neural networks (**NCAT** (Zhuang et al., 2022), **GMOCAT** (Wang et al., 2023a)); and a gradient-approximation heuristic approach (**BECAT**) (Zhuang et al., 2023; 2025).

## 5.3. Results and Discussion

In this section, we compare the performance on two classic human testing tasks introduced above to evaluate the effectiveness of the proposed GPM framework.

**Task 1: Simulation of Ability Estimation.** The results in Figure 4(a) illustrate the MSE trends of different methods on IRT. As the size of the selected question subset increases, the MSE for all methods decreases. This aligns with a common trend in parameter estimation: larger sample sizes provide more information, thereby reducing estimation error. In the early stages of the test ($t = 5$), when the available data is minimal, the differences between methods are relatively small. However, GPM shows slightly faster error reduction during this phase. As the number of selected questions grows, GPM effectively reduces the error at a faster rate and consistently achieves significantly lower estimation errors throughout the testing process. Notably, compared to the implicit methods, such as FSI, the proposed GPM achieves the same estimation error while requiring up to 20% fewer questions. On average, GPM reduces the required test length by 9% to reach the same estimation accuracy, demonstrating its efficiency in ability estimation and its potential to shorten tests without sacrificing accuracy.

**Task 2: Examinee Score Prediction.** In Table 1, we use a standard IRT model to estimate examinee ability. Among all baselines, GPM consistently achieves the highest performance across all datasets and subset sizes. For example, on the EXAM dataset, GPM outperforms the best baseline (UATS) by +2.7% AUC at step 20. Notably, GPM surpasses reinforcement learning and meta learning methods

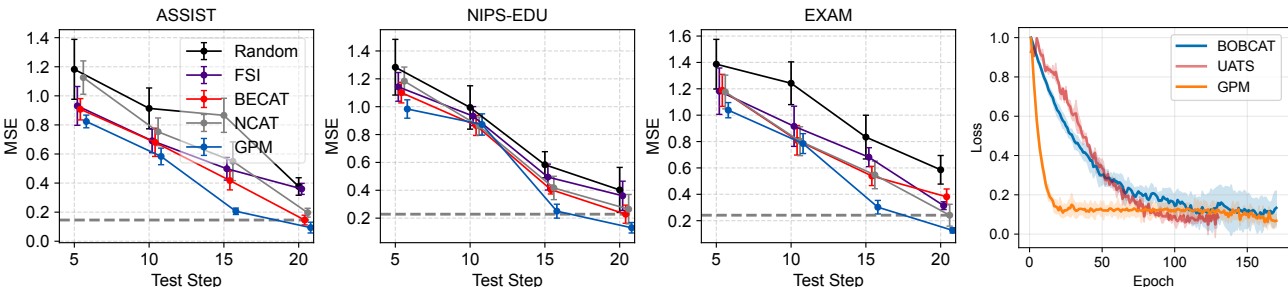

*Figure 4.* (a) Simulation results for ability estimation: MSE of ability estimation, $\mathbb{E}\|\theta^t - \theta_0\|^2$, under different subset sizes (steps) for five representative question selection algorithms. Results are averaged over 10 repetitions, with error bars indicating the standard deviation. (b) Loss convergence curves of different data-driven methods on the ASSIST. Non data-driven baselines (e.g., BECAT) do not require training and are therefore omitted.

*Table 1.* The performances on Examinee Score Prediction at different test steps (subset size). It reports both ACC and AUC metrics at various steps (5, 10, and 20), based on the IRT user model for ability estimation (results using NeuralCDM are provided in Appendix). All results are averaged over 10 independent runs with different seeds. Bold values indicate statistically significant improvements over the best baseline according to a paired two-sided t-test (p-value $< 0.01$).

| Method | ASSIST (ACC/AUC) | | | NIPS-EDU (ACC/AUC) | | | EXAM (ACC/AUC) | | |
|---|---|---|---|---|---|---|---|---|---|
| | @5 | @10 | @20 | @5 | @10 | @20 | @5 | @10 | @20 |
| Random | 71.01/70.68 | 72.20/71.91 | 73.07/72.61 | 66.45/69.05 | 68.23/71.66 | 70.23/74.82 | 77.58/70.34 | 78.59/71.91 | 80.40/74.22 |
| FSI | 71.77/71.33 | 72.94/72.48 | 73.24/73.54 | 67.70/70.60 | 69.62/73.62 | 71.03/76.24 | 77.37/70.57 | 78.79/72.21 | 81.01/74.89 |
| KLI | 71.93/71.38 | 72.73/72.52 | 73.17/73.57 | 67.09/69.79 | 69.27/73.30 | 70.42/75.73 | 77.37/70.57 | 78.79/72.21 | 81.01/74.70 |
| MAAT | 72.20/71.54 | 72.33/72.58 | 73.22/73.08 | 66.70/70.32 | 69.13/72.41 | 69.07/74.46 | 76.97/70.38 | 78.79/72.12 | 80.61/74.65 |
| BOBCAT | 72.31/71.68 | 72.36/72.28 | 73.70/73.39 | 69.51/74.42 | 70.94/75.73 | 71.73/76.58 | 80.81/68.17 | 83.84/72.04 | 83.43/72.88 |
| NCAT | 72.28/71.53 | 72.55/72.31 | 73.81/73.50 | 67.30/72.11 | 70.68/75.80 | 71.91/76.66 | 80.92/70.72 | 83.99/72.71 | 84.02/74.29 |
| UATS | 72.58/72.81 | 72.09/72.71 | 74.27/74.82 | 67.28/73.53 | 70.55/74.89 | 71.81/76.50 | 79.22/70.87 | 82.32/73.28 | 84.92/75.20 |
| BECAT | 71.92/71.44 | 73.01/72.73 | 73.96/73.61 | 66.98/73.15 | 71.61/75.87 | 72.00/76.82 | 80.99/70.74 | 83.85/72.88 | 84.29/75.00 |
| **GPM** | **72.82/73.58** | **73.87/73.76** | **75.01/75.27** | **69.68/74.67** | **72.27/76.16** | **73.98/78.10** | **81.19/71.09** | **84.19/73.99** | **86.02/77.89** |

like BOBCAT and NCAT, demonstrating that optimizing for gradient alignment is more effective than handcrafted selection criteria. These results highlight: First, GPM surpasses latest SOTA deep learning subset selection methods like UATS and NCAT. This suggests that accurately approximating the gradient of the ability estimation objective is more critical than modeling complex examinee-question interactions. Second, by using gradient alignment as a supervisory signal, GPM can outperform BECAT, which attempts to approximate the entire parameter space $\Theta$, achieving better generalization and cold-start performance, surpassing other data-driven methods by approximately 2%.

### 5.4. Convergence Comparison

Here, we compare the convergence speed and stability of different data-driven methods during training. Figure 4(b) illustrates the loss curves of three methods. The solid lines represent the mean loss over 5 training runs, while the shaded regions indicate the standard deviation. GPM demonstrates faster convergence and maintains stability throughout the training process. In contrast, other methods exhibit unstable convergence and high variability, reflecting their sensitivity

*Table 2.* Score prediction performance comparison (AUC) of 4 different data-driven methods under varying conditions on the EXAM dataset. Bold values indicate statistically significant improvements over the best baseline according to a paired two-sided t-test (p-value $< 0.01$). The full results are in Appendix

| Conditions | GMOCAT | BOBCAT | UATS | **GPM** |
|---|---|---|---|---|
| **Original** | $74.33 \pm 0.28$ | $72.88 \pm 0.31$ | $73.45 \pm 0.27$ | **$77.89 \pm 0.28$** |
| **90% Data** | $74.28 \pm 0.30$ | $72.01 \pm 0.33$ | $73.13 \pm 0.29$ | **$76.11 \pm 0.21$** |
| **80% Data** | $73.71 \pm 0.32$ | $71.89 \pm 0.35$ | $72.90 \pm 0.31$ | **$75.09 \pm 0.23$** |
| **50% Data** | $72.85 \pm 0.34$ | $70.65 \pm 0.38$ | $72.71 \pm 0.33$ | **$74.21 \pm 0.25$** |
| **5% Noise** | $73.16 \pm 0.29$ | $72.00 \pm 0.32$ | $73.12 \pm 0.28$ | **$76.00 \pm 0.20$** |
| **10% Noise** | $73.08 \pm 0.31$ | $71.78 \pm 0.34$ | $72.87 \pm 0.30$ | **$74.79 \pm 0.29$** |
| **15% Noise** | $72.50 \pm 0.33$ | $71.69 \pm 0.36$ | $71.43 \pm 0.32$ | **$73.30 \pm 0.32$** |

to initialization and optimization dynamics. Compared to other meta learning frameworks, e.g., BOBCAT and UATS, GPM effectively leverages the gradient information from each step of the inner-loop optimization for ability estimation as supervision. This eliminates the need to wait for the inner-loop training to fully converge before optimizing the outer-loop $S_\phi$. These results validate the training efficiency of GPM and serve as empirical support for the theoretical convergence guarantees in Theorem 4.3.

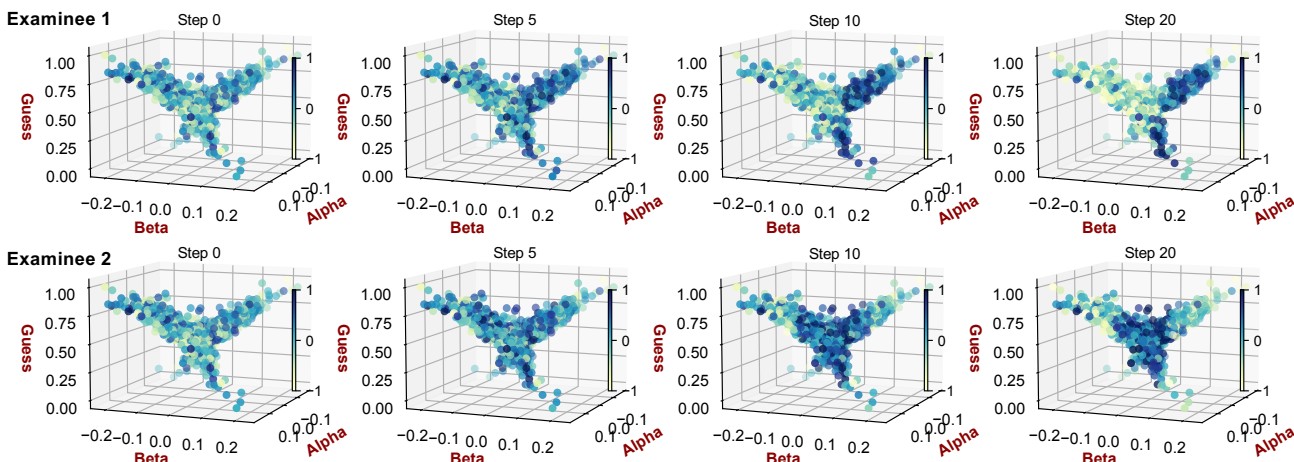

*Figure 5.* The characteristics of all questions in the ASSIST dataset, visualized based on their IRT parameters: difficulty ($\beta$), discrimination ($\alpha$), and guessing factor ($c$). The shading intensity represents the logits of a question being selected under the GPM for two different examinees during the testing process.

### 5.5. Robustness

In real-world testing scenarios, examinee response data is typically limited in scale and subject to significant noise (Zhuang et al., 2022). Specifically, 1) the amount of response data available for training is limited. This includes both the number of examinees and the number of responses per examinee; 2) examinee responses are inherently uncertain due to factors such as *guessing* (correctly answering questions they should have gotten wrong) and *slipping* (incorrectly answering questions they should have gotten right). These behaviors can result in response labels ($y$) that do not accurately reflect an examinee's true ability.

To evaluate the robustness of the proposed methods under these realistic constraints, we conduct experiments under two challenging conditions: (1) reducing the size of the training dataset and (2) introducing noise into the data by flipping response labels ($y$) with varying probabilities. Table 6 summarizes the performance degradation under these conditions. The results show that GPM maintains high predictive accuracy even under sparse data conditions, whereas other methods get significant performance drops. Furthermore, even under high-noise conditions (e.g., 15% label noise), GPM demonstrates superior robustness, consistently outperforming other methods. These results highlight its resilience in real-world testing scenarios, where data limitations and noise are unavoidable.

### 5.6. Characteristics of Selected Questions

To investigate the characteristics of the questions selected by GPM, we visualize them during the testing process. Figure 5 illustrates the distribution of all questions in pool $Q$ across the three key IRT parameters: difficulty, discrimination, and guessing factor. The shading intensity represents

the logits of a question being selected. As the test progresses and more examinee response data is collected, the range of selectable questions $S$ gradually narrows. Specifically, at the beginning of the test, GPM tends to select questions that are more evenly distributed across the pool. As the test progresses, GPM adapts to each examinee, selecting questions that are better suited to their estimated ability. For example, the range of selectable questions in terms of difficulty ($\beta$) contracts to different regions of the pool depending on examinee's ability. Additionally, GPM demonstrates a preference for questions with high discrimination ($\alpha$), aligning with the principles of well-known Fisher Information. The visualization highlights GPM's ability to identify the informative questions, and its adaptive and personalized nature.

## 6. Conclusion and Future Work

Adaptive testing addresses a fundamental subset selection problem: selecting a small but informative subset of questions from a large question pool to accurately estimate an examinee's ability. In this work, we proposed GPM (Gradient Path Matching), a simple yet effective framework that learns to approximate gradient signals directly from data. Rather than matching gradients globally across the entire ability space, GPM focuses on aligning gradients along the possible optimization path toward the final estimate. It can significantly reduce computational cost while maintaining high estimation accuracy. It shows great accuracy, convergence, and robustness under challenging conditions, e.g., data sparsity, high noise, and ranking-based evaluation tasks.

Instead of relying on complex model structures, this study takes a simpler, more fundamental approach, focusing on the core challenges of adaptive testing. However, this line of research opens up several promising directions for future

work. For example, incorporating richer information (e.g., question tags, textual data, and examinee profiles), and leveraging advanced network architectures (particularly for the question selection network). We hope this work provides a foundation or a perspective for the future design of scalable and reliable adaptive testing systems.

## Impact Statement

Adaptive testing research originated in the mid-20th century and has evolved over more than seventy years (Lord, 1952; William, 1979). In human assessment, adaptive testing has been widely adopted in high stakes examinations. Although early concerns were raised regarding fairness (e.g., different examinees receiving different questions), advances in intelligent assessment and online education have led to broad acceptance. This is largely because adaptive testing evaluates examinees based on question characteristics (e.g., difficulty) rather than specific questions, which ensures the comparability of ability estimates. In practice, equating and scaling techniques are commonly applied, as in standardized examinations such as the GRE. To accommodate diverse scenarios, this work considers both personalized and non personalized adaptive testing settings. It allows practitioners to select appropriate strategies depending on assessment requirements

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

# A. Algorithm of Personalized GPM

---

**Algorithm 2** Personalized Gradient Path Matching (Personalized GPM)

---

**Require:** Response data $\{\tau\}$, training steps $N$, test length $T$, inner-loop steps $K$, learning rates $\eta_\theta, \eta_\phi$
1: Initialize selection model parameters $\phi$
2: **for** $n = 1, 2, \ldots, N$ **do**
3:      Sample one examinee's full response record $\tau = \{(q_i, y_i)\}_{i \in Q}$
4:      Initialize response history $\tau_0 = \emptyset$
5:      **for** $t = 1, 2, \ldots, T$ **do**
6:          Compute question selection probabilities: $S_t = S(\tau_{t-1}; \phi)$.
7:          Sample next question: $q_t \sim \text{GumbelSoftmax}(S_t)$.
8:          Obtain response $y_t$ and update history: $\tau_t = \tau_{t-1} \cup \{(q_t, y_t)\}$.
9:          Initialize ability parameter $\theta^{(0)}$
10:         **for** $k = 1, 2, \ldots, K$ **do**
11:            Update ability parameters on selected questions: $\theta^{(k)} = \theta^{(k-1)} - \eta_\theta \nabla_{S_t}(\theta^{(k-1)})$.
12:         **end for**
13:         Compute gradient discrepancy: $\Delta(S_t, \theta^{(K)}) = \left\| \nabla_{S_t}(\theta^{(K)}) - \nabla_Q(\theta^{(K)}) \right\|$.
14:         Update policy parameters: $\phi = \phi - \eta_\phi \nabla_\phi \Delta(S_t, \theta^{(K)})$.
15:      **end for**
16: **end for**
17: **Output:** Trained personalized selection model $S(\cdot; \phi)$

---

# B. Proofs of Lemma 3.2

**Lemma B.1** (Gradient Approximation Bounds Estimation Error). *Let $\theta_S$ and $\theta^*$ denote the estimated ability parameters based on a subset $S \subseteq Q$ and the full set $Q$, respectively. When using IRT to estimate examinee ability, there exists a constant $C > 0$ such that the estimation error is bounded by the maximum gradient approximation error:*

$$\min_S \|\theta_S - \theta^*\| \le C \min_S \sup_{\theta \in \Theta} \left\| \sum_{j \in S} \gamma_j \nabla \ell(q_j, y_j; \theta) - \sum_{i \in Q} \nabla \ell(q_i, y_i; \theta) \right\|$$

*Proof.* Let the empirical risk of ability estimation over the full question pool be defined as:

$$L_Q(\theta) := \sum_{i \in Q} \ell_i(q_i, y_i; \theta), \quad \theta^* := \arg\min_{\theta \in \Theta} L_Q(\theta).$$

Let the weighted empirical risk over a subset $S \subseteq Q$ be defined as:

$$L_S(\theta) := \sum_{j \in S} \gamma_j \ell_j(q_j, y_j; \theta), \quad \theta_S := \arg\min_{\theta \in \Theta} L_S(\theta).$$

When using L2-norm IRT for ability estimation, the loss function $\ell$ is $\mu$-strongly convex and differentiable. Then, $L_Q$ is $|Q|\mu$-strongly convex, and $L_S$ is $\mu \sum_{j \in S} \gamma_j$-strongly convex (Yu et al., 2024a; Zhuang et al., 2023). Since $\theta^*$ and $\theta_S$ are the minimizers of $L_Q$ and $L_S$, respectively, we have: $\nabla L_Q(\theta^*) = 0$ and $\nabla L_S(\theta_S) = 0$. we apply the standard inequality:

$$L_S(\theta^*) \ge L_S(\theta_S) + \nabla_S(\theta_S)^\top (\theta^* - \theta_S) + \frac{\mu \sum_{j \in S} \gamma_j}{2} \|\theta^* - \theta_S\|^2,$$

$$L_S(\theta_S) \ge L_S(\theta^*) + \nabla_S(\theta^*)^\top (\theta_S - \theta^*) + \frac{\mu \sum_{j \in S} \gamma_j}{2} \|\theta_S - \theta^*\|^2.$$

Adding the two inequalities and applying Cauchy–Schwarz inequality, we obtain:

$$\mu \sum_{j \in S} \gamma_j \|\theta_S - \theta^*\|^2 \le |\langle \nabla_S(\theta^*) - \nabla_Q(\theta^*), \theta_S - \theta^* \rangle| \le \|\nabla_S(\theta^*) - \nabla_Q(\theta^*)\| \|\theta_S - \theta^*\|.$$

Thus

$$\|\theta_S - \theta^*\| \leq \frac{1}{\mu \sum_{j \in S} \gamma_j} \|\nabla_S(\theta^*) - \nabla_Q(\theta^*)\|.$$

Since the inequality holds pointwise at $\theta^*$, it also holds for the worst-case over $\theta \in \Theta$:

$$\|\theta_S - \theta^*\| \leq \frac{1}{\mu \sum_{j \in S} \gamma_j} \sup_{\theta \in \Theta} \|\nabla_S(\theta) - \nabla_Q(\theta)\|.$$

Taking the minimum over all subsets $S \subseteq Q$ of size $T$, we obtain:

$$\min_S \|\theta_S - \theta^*\| \leq C \cdot \min_S \sup_{\theta \in \Theta} \|\nabla_S(\theta) - \nabla_Q(\theta)\|$$

where $C = \frac{1}{\mu \sum_{j \in S} \gamma_j}$. This completes the proof. □

## C. Proofs of Lemma 4.1

**Lemma C.1.** *Let $\{\theta_S^{(k)}\}_{k=1}^K \subset \Theta$ be a set of ability estimates such that for every $\theta \in \Theta$, there exists some $k \in \{1, \ldots, K\}$ with $\|\theta - \theta_S^{(k)}\| \leq \varepsilon$ and $K \geq 1$. When using IRT for ability estimation, there exists a constant $L$ such that:* $\min_S \sup_{\theta \in \Theta} \|\nabla_S(\theta) - \nabla_Q(\theta)\| \leq \min_S \sum_{k=1}^K \|\nabla_S(\theta_S^{(k)}) - \nabla_Q(\theta_S^{(k)})\| + L\varepsilon.$

*Proof.* Let us define the pointwise gradient discrepancy function $\Delta(S, \theta) := \|\nabla_S(\theta) - \nabla_Q(\theta)\|$. According to Lemma 4.2, $\Delta(S, \theta)$ is $L$-Lipschitz continuous in $\theta$, i.e., for all $\theta, \theta' \in \Theta$,

$$|\Delta(S, \theta) - \Delta(S, \theta')| \leq L \cdot \|\theta - \theta'\|.$$

Now, consider any $\theta \in \Theta$. There exists some $\theta_S^{(k)}$ such that $\|\theta - \theta_S^{(k)}\| \leq \varepsilon$. Then:

$$\Delta(S, \theta) \leq \Delta(S, \theta_S^{(k)}) + L\|\theta - \theta_S^{(k)}\| \leq \Delta(S, \theta_S^{(k)}) + L\varepsilon.$$

Taking the supremum over all $\theta \in \Theta$, we have:

$$\sup_{\theta \in \Theta} \Delta(S, \theta) \leq \max_{k=1,\ldots,K} \Delta(S, \theta_S^{(k)}) + L\varepsilon.$$

Next, observe that $\max_k \Delta(S, \theta_S^{(k)}) \leq \sum_{k=1}^K f_S(\theta_S^{(k)})$, since the maximum is upper bounded by the sum. Combining the two inequalities, we obtain:

$$\sup_{\theta \in \Theta} \Delta(S, \theta) \leq \sum_{k=1}^K \Delta(S, \theta_S^{(k)}) + L\varepsilon.$$

Taking the minimum over all subsets $S \subseteq Q$ on both sides gives:

$$\min_S \sup_{\theta \in \Theta} \|\nabla_S(\theta) - \nabla_Q(\theta)\| \leq \min_S \sum_{k=1}^K \left\|\nabla_S(\theta_S^{(k)}) - \nabla_Q(\theta_S^{(k)})\right\| + L\varepsilon.$$

This completes the proof. □

## D. Proofs of Lemma 4.2

**Lemma D.1** (A Lipschitz Continuous Gradient). *The function of gradient sequence discrepancy $\Delta(S, \theta_S)$ serves as the loss function for optimizing $S$. When using IRT for estimating ability $\theta$, $\Delta(S, \theta)$ is Lipschitz continuous, and gradient $\nabla_S \Delta(S, \theta_S)$ is also Lipschitz with constant $L_\Delta$. Then, for any $S, S'$:*

$$\|\nabla_S \Delta(S, \theta_S) - \nabla_S \Delta(S', \theta_{S'})\| \leq L_\Delta \|S - S'\|. \tag{11}$$

*Proof.* The ability estimation function in IRT is strongly convex ($\mu$-strong) w.r.t. ability $\theta$. The cross-entropy loss function based on IRT ability estimation and its derivative norms are bounded (Lipschitz continuous). Specifically, for the gradient functions $g$ over $S$ and $Q$, there exist constants $L_S$ and $L_Q$ such that for any $\theta$ and $\theta'$,

$$\|\nabla_S(\theta) - \nabla_S(\theta')\| \le L_S\|\theta - \theta'\|,$$
$$\|\nabla_Q(\theta) - \nabla_Q(\theta')\| \le L_Q\|\theta - \theta'\|.$$

The objective function for optimizing subset $S$ is: $\Delta(S, \theta) = \|\nabla_S(\theta) - \nabla_Q(\theta)\|$. It is found that its gradient with respect to $\theta$, $\nabla_\theta\Delta(S, \theta)$, is Lipschitz continuous. Using the reverse triangle inequality, for any $\theta$ and $\theta'$,

$$
\begin{aligned}
\|\Delta(S, \theta) - \Delta(S, \theta')\| =& |\|\nabla_S(\theta) - \nabla_Q(\theta)\| - \|\nabla_S(\theta') - \nabla_Q(\theta')\|| \\
\le& \|(\nabla_S(\theta) - \nabla_Q(\theta)) - (\nabla_S(\theta') - \nabla_Q(\theta'))\| \\
\le& \|\nabla_S(\theta) - \nabla_S(\theta')\| + \|\nabla_Q(\theta) - \nabla_Q(\theta')\| \\
\le& (L_S + L_Q)\|\theta - \theta'\|.
\end{aligned}
\tag{12}
$$

Thus, $\Delta(S, \theta)$ with respect to $\theta$ is Lipschitz continuous with constant $L_S + L_Q$.

Next, assume $S \in \mathbb{R}^{|Q|}$ is a non-negative continuous vector (representing the probability of selecting each item in $Q$, i.e., the item's weight), and $\|S\|_1 = \sum \gamma = |Q|$:

$$\|\nabla_S\Delta(S, \theta)\| = \|\nabla_Q(\theta)\| \le \|\sum_{i \in Q} \nabla\ell_i(q_i, y_i; \theta)\| \le |Q|\sigma_l$$

Thus, $\Delta(S, \theta)$ with respect to subset $S$ is also Lipschitz continuous with constant $|Q|\sigma_l$.

Similarly, to prove that the gradient $\nabla\Delta(S, \theta_S^{(k)})$ is also Lipschitz continuous, $\nabla_S^2\Delta(S, \theta)$, $\nabla_\theta\nabla_S\Delta(S, \theta)$, $\nabla_S\nabla_\theta\Delta(S, \theta)$, and $\nabla_\theta^2\Delta(S, \theta)$ are all bounded (Yu et al., 2024a). Therefore, the Jacobian matrix of $\nabla\Delta(S, \theta)$ is bounded and the domain formed by $\theta$ and the vector $S$ in IRT is compact, making $\nabla\Delta(S, \theta)$ Lipschitz continuous.

Based on the conclusion from Lemma 2.2 in (Ghadimi & Wang, 2018), the gradient $\nabla\Delta(S, \theta_S)$ with respect to $S$ is $L_\Delta$-Lipschitz continuous:

$$\|\nabla_S\Delta(S, \theta_S) - \nabla_S\Delta(S', \theta_{S'})\| \le L_\Delta\|S - S'\|. \tag{13}$$

This completes the proof. $\qquad\square$

## E. Proofs of Theorem 4.3

**Theorem E.1** (Upper Bound of Gradient Norm). *Define the parameter $L_\Delta$ as in Lemma 4.2. When optimizing $S$ with a learning rate $\eta_s < \frac{2}{L_\Delta}$ and the ability parameter $\theta$ is optimized to $\theta^{(k)}$, the gradient discrepancy loss $\Delta(S, \theta_S)$ over $N$ steps satisfies:*

$$\frac{1}{N}\sum_{n=1}^{N}\left\|\nabla_S\Delta\left(S^{(n)}, \theta_{S^{(n)}}^{(k)}\right)\right\|^2 \le \frac{2\Delta\left(S^{(0)}, \theta_{S^{(0)}}^{(k)}\right)}{N(2\eta_s - L_\Delta\eta_s^2)}. \tag{14}$$

*Proof.* Let $\Delta(S) = \Delta(S, \theta_S^{(k)})$ denote the gradient discrepancy loss at the optimal value $\theta^{(k)}$. Based on Lemma 2, the gradient $\nabla_S\Delta(S)$ is Lipschitz continuous. Therefore, we have:

$$
\begin{aligned}
\Delta(S^{(n+1)}) \le& \Delta(S^{(n)}) + \langle\nabla\Delta(S^{(n)}), S^{n+1} - S^{(n)}\rangle + \frac{L_\Delta}{2}\|S^{(n+1)} - S^{(n)}\|^2 \\
=& \Delta(S^{(n)}) - \eta_s\|\nabla\Delta(S^{(n)})\|^2 + \frac{L_\Delta\eta_s^2}{2}\|\nabla\Delta(S^{(n)})\|^2.
\end{aligned}
\tag{15}
$$

Thus, we have:

$$\|\nabla\Delta(S^{(n)})\|^2 \le \frac{2}{2\eta_s - L_\Delta\eta_s^2}(\Delta(S^{(n)}) - \Delta(S^{(n+1)})). \tag{16}$$

Summing from $n = 0$ to $N - 1$, we obtain:

$$\frac{1}{N} \sum_{n=0}^{N-1} \|\nabla \Delta(S^{(n)})\|^2 \leq \frac{2}{N(2\eta_s - L_\Delta \eta_s^2)} \left(\Delta(S^{(0)}) - \Delta(S^{(N)})\right) \leq \frac{2\Delta(S^{(0)})}{N(2\eta_s - L_\Delta \eta_s^2)}. \tag{17}$$

This completes the proof.

$\square$

# F. Additional Experiments

**Ability Ranking Experiment**   While the primary goal of adaptive testing is accurate ability estimation, certain testing scenarios, such as selective exams or ranking-based evaluations, prioritize the consistency of ability rankings (i.e., the partial order of ability values) rather than absolute accuracy. Prior research (Liu et al., 2024b) has shown that many methods achieving high accuracy in ability estimation often fail to produce consistent rankings. For instance, as shown in Table 4, methods like BOBCAT and UATS demonstrate suboptimal ranking performance. To explore this issue, we follow the experimental setup in (Liu et al., 2024b) and compute the Kendall rank correlation coefficient (Kendall, 1938) across all estimated ability values to evaluate whether BEAT framework can consistently rank examinees with varying ability levels. As shown in Table 4, GPM achieves higher overall ranking consistency compared to other algorithms when examinees answer the same number of questions. However, it does not yet fully surpass CCAT, a method specifically designed to optimize for ranking consistency. This suggests that GPM, despite being explicitly designed to optimize for accurate ability estimation rather than ranking, still performs competitively in ranking tasks. This highlights the effectiveness of GPM in ranking scenarios.

*Table 3.* Statistics of the datasets

| Dataset | ASSIST | NIPS-EDU | EXAM |
|---|---|---|---|
| #Examinees | 20,704 | 220,274 | 9,214 |
| #Questions | 15,071 | 27,613 | 1,650 |
| #Response logs | 1,768,253 | 19,181,192 | 133,398 |
| #Response logs per examinee | 85.41 | 87.08 | 14.48 |
| #Response logs per question | 117.33 | 694.64 | 80.85 |

*Table 4.* The ranking consistency of different methods is evaluated using the Kendall rank correlation coefficient. Higher values indicate better consistency in ranking examinees' abilities. The results are reported under the same number of answered questions on the EXAM dataset using IRT. Note that CCAT (Liu et al., 2024b) is a question selection algorithm specifically designed to optimize ranking consistency. Paired t-tests are conducted and statistically significant differences ($p < 0.01$) are indicated with an asterisk (*). The best-performing results are shown in bold.

| Metric@Step | Random | FSI | BOBCAT | UATS | NCAT | GMOCAT | CCAT | BEAT | GPM |
|---|---|---|---|---|---|---|---|---|---|
| **Kendall@5** | $0.673_{\pm0.051}$ | $0.712_{\pm0.035}$ | $0.596_{\pm0.013}$ | $0.743_{\pm0.031}$ | $0.761_{\pm0.024}$ | $0.733_{\pm0.042}$ | $0.783_{\pm0.030}$ | $0.753_{\pm0.032}$ | $\mathbf{0.792_{\pm0.020}}$* |
| **Kendall@10** | $0.712_{\pm0.044}$ | $0.777_{\pm0.023}$ | $0.689_{\pm0.014}$ | $0.774_{\pm0.014}$ | $0.805_{\pm0.014}$ | $0.815_{\pm0.014}$ | $0.830_{\pm0.031}$ | $0.832_{\pm0.024}$ | $\mathbf{0.854_{\pm0.012}}$* |
| **Kendall@15** | $0.757_{\pm0.042}$ | $0.815_{\pm0.028}$ | $0.723_{\pm0.012}$ | $0.781_{\pm0.019}$ | $0.826_{\pm0.012}$ | $0.835_{\pm0.017}$ | $\mathbf{0.878_{\pm0.022}}$* | $0.841_{\pm0.021}$ | $0.869_{\pm0.015}$ |
| **Kendall@20** | $0.770_{\pm0.026}$ | $0.825_{\pm0.010}$ | $0.753_{\pm0.015}$ | $0.793_{\pm0.011}$ | $0.843_{\pm0.005}$ | $0.858_{\pm0.003}$ | $0.881_{\pm0.024}$ | $0.868_{\pm0.018}$ | $\mathbf{0.899_{\pm0.015}}$ |
| **Average** | $0.728_{\pm0.041}$ | $0.782_{\pm0.024}$ | $0.690_{\pm0.014}$ | $0.773_{\pm0.019}$ | $0.809_{\pm0.014}$ | $0.810_{\pm0.019}$ | $0.843_{\pm0.027}$ | $0.823_{\pm0.024}$ | $\mathbf{0.854_{\pm0.016}}$ |

**Examinee Score Prediction on NeuralCDM**   Table 5 reports both ACC and AUC metrics at various steps (5, 10, and 20), based on neural network-based model (NeuralCDM) for ability estimation. NeuralCDM can cover many IRT and cognitive diagnosis models, such as MIRT (Reckase, 2009) and MF (Desmarais, 2012). Note that information/uncertainty-based selection algorithms (e.g., FSI) cannot be applied to deep learning methods, which is why these methods are excluded from the results in Table 5.

*Table 5.* The performances on Examinee Score Prediction at different test steps (subset size) on NeuralCDM. Bold values indicate statistically significant improvements (p-value < 0.01) over the best baseline.

| Method | ASSIST (ACC/AUC) | | | NIPS-EDU (ACC/AUC) | | | EXAM (ACC/AUC) | | |
|---|---|---|---|---|---|---|---|---|---|
| | @5 | @10 | @20 | @5 | @10 | @20 | @5 | @10 | @20 |
| Random | 71.52/71.19 | 72.66/72.06 | 72.67/72.83 | 67.19/69.32 | 68.44/71.56 | 70.57/74.99 | 79.80/72.58 | 79.80/74.81 | 79.80/78.40 |
| MAAT | 72.36/70.98 | 72.52/72.33 | 71.74/72.27 | 67.86/70.12 | 70.07/72.58 | 70.66/75.83 | 82.82/70.32 | 82.83/74.11 | 83.82/79.44 |
| BOBCAT | 72.69/71.45 | 72.89/72.84 | 73.87/72.84 | 71.13/76.00 | 72.52/77.87 | 73.47/79.00 | 78.18/78.24 | 78.19/81.47 | 78.18/79.49 |
| NCAT | 72.28/71.59 | 72.63/72.37 | 73.90/73.59 | 70.47/74.10 | 72.81/77.99 | 73.47/79.12 | 82.30/78.77 | 83.19/81.47 | 81.53/79.49 |
| UATS | 73.13/72.34 | 72.93/73.14 | 73.14/72.88 | 71.99/75.24 | 73.19/78.08 | 74.10/79.77 | 81.30/77.19 | 82.40/80.98 | 83.78/80.86 |
| BECAT | 72.30/71.60 | 73.11/72.97 | 74.13/73.70 | 71.33/76.30 | 73.09/78.34 | 73.58/79.36 | 82.84/78.75 | 83.22/81.49 | 84.77/79.70 |
| **GPM** | **74.44/72.89** | **73.55/73.78** | **74.56/74.32** | **72.23/77.13** | **74.34/79.55** | **74.76/81.66** | **83.72/80.11** | **84.10/82.90** | **85.11/81.78** |

*Table 6.* Score prediction performance comparison (AUC) of 4 different data-driven methods under varying conditions, including original data (no reduction or noise), reduced training dataset sizes (-10%, -20%, -50%), and response label perturbation levels (5%, 10%, 15%). Label perturbation refers to flipping the binary labels ($y$) with a certain probability. Bold values indicate statistical significance (p-value < 0.01).

| Conditions | ASSIST | | | | NIPS-EDU | | | | EXAM | | | |
|---|---|---|---|---|---|---|---|---|---|---|---|---|
| | GMOCAT | BOBCAT | UATS | GPM | GMOCAT | BOBCAT | UATS | GPM | GMOCAT | BOBCAT | UATS | GPM |
| **Original** | 73.53 | 73.39 | 74.82 | **75.27** | 76.61 | 76.58 | 76.50 | **78.10** | 74.33 | 72.88 | 73.45 | **77.89** |
| **90% Data** | 73.44 | 73.28 | 73.37 | **74.85** | 75.72 | 76.32 | 75.58 | **77.27** | 74.28 | 72.01 | 73.13 | **76.11** |
| **80% Data** | 72.10 | 73.21 | 73.27 | **74.27** | 74.87 | 76.00 | 75.23 | **76.86** | 73.71 | 71.89 | 72.90 | **75.09** |
| **50% Data** | 71.98 | 73.02 | 73.10 | **73.89** | 73.82 | 75.23 | 74.82 | **76.10** | 72.85 | 70.65 | 72.71 | **74.21** |
| **5% Noise** | 73.48 | 73.38 | 72.63 | **74.01** | 75.98 | 75.82 | 76.27 | **77.44** | 73.16 | 72.00 | 73.12 | **76.00** |
| **10% Noise** | 72.22 | 73.11 | 71.93 | **73.52** | 75.12 | 74.17 | 75.17 | **76.98** | 73.08 | 71.78 | 72.87 | **74.79** |
| **15% Noise** | 71.87 | 72.01 | 71.92 | **72.97** | 73.42 | 74.16 | 74.99 | **75.31** | 72.50 | 71.69 | 71.43 | **73.30** |

