# OpenReview forum: "Efficient Adaptive Testing via Gradient Path Matching Subset Selection for AI Education"
_ICML.cc/2026/Conference — ICML 2026 regular_

### Official Review · Reviewer_xwCv · 2026-03-11

**Soundness:** 2
**Presentation:** 3
**Significance:** 3
**Originality:** 2
**Overall Recommendation:** 3
**Confidence:** 4

**Summary:**

This paper proposes GPM, an efficient adaptive testing framework that integrates trajectory-based gradient matching with a sequential decision-making process for item selection. Specifically, the method aligns the gradients of selected question subsets with those of the full item pool along the optimization paths of ability estimation, rather than seeking global alignment across the entire parameter space. To facilitate this, a meta-learning-based predictor is introduced to provide intermediate gradient supervision, allowing the system to learn an explicit selection policy from large-scale historical data. Extensive experiments on multiple benchmark educational datasets are conducted to validate the efficiency and estimation accuracy of the proposed framework.

**Compliance With Llm Reviewing Policy:**

Affirmed.

**Key Questions For Authors:**

1. **How does the path approximation error vary across training stages and different ability initializations?**

	The paper provides a theoretical bound for the approximation error in Lemma 4.1 and proves convergence in Theorem 4.3, both of which heavily depend on the constant $\epsilon$. However, the manuscript lacks an empirical analysis of how this error fluctuates during different phases of the bilevel optimization process (e.g., early vs. late training). Furthermore, it remains unclear whether the learned selection policy is robust to biased initializations of the ability parameter $\theta^{(0)}$, which could lead the optimization trajectory away from the "true" gradient path. Providing empirical evidence on the stability of the error bound and discussing the model's robustness under diverse initializations would significantly strengthen the theoretical justification of the trajectory-based relaxation.

2. **What are the specific computational costs associated with the training phase of GPM compared to existing baselines?**

	The paper highlights the efficiency of GPM during the inference/testing stage, noting a complexity of $O(|Q|)$. However, the training phase involves a bilevel optimization framework that is traditionally known for being computationally expensive and memory-intensive. Currently, the paper does not provide explicit comparisons regarding wall-clock time or GPU memory usage against state-of-the-art methods such as reinforcement learning-based approaches or global gradient matching frameworks (e.g., GMOCAT). Including a detailed breakdown of the training overhead and a scalability analysis for large-scale question pools would help quantify the practical benefits and industrial feasibility of the proposed method.

3. **How sensitive is the model performance to the trajectory length $K$ and the historical path sampling strategy?**

	The effectiveness of the GPM framework relies on approximating the gradient path using $K$ optimization steps. The paper does not clearly explore the sensitivity of the ability estimation accuracy to the choice of $K$, nor does it identify a potential point of diminishing returns where increasing $K$ no longer yields significant performance gains. Additionally, since the selection algorithm is trained on paths sampled from historical data, it is unclear how the quality or "representativeness" of these trajectories affects the learned selector's generalization. Clarifying the impact of $K$ through additional ablation studies and discussing how the trajectory sampling strategy influences the model's stability would provide a more comprehensive understanding of the framework's data dependency.

**Limitations:**

While the proposed GPM framework demonstrates promising improvements in efficiency by aligning gradients along optimization paths rather than the entire ability space, several limitations remain.First, the theoretical analysis relies on the assumption that the path approximation error, bounded by the constant $\epsilon$ in Lemma 4.1, remains sufficiently small throughout the training process. Although this assumption simplifies the convergence analysis, it may not fully reflect the complex optimization dynamics encountered in practice, especially when the initial ability estimate $\theta^{(0)}$ is highly inaccurate, which could lead the model to follow unstable or unrepresentative training trajectories.Second, the method introduces a bilevel optimization framework that, while efficient during inference, incurs substantial computational and memory overhead during the training phase. The effectiveness of the gradient matching process is sensitive to hyperparameters such as the number of optimization steps ($K$) and the meta-learning rate. As a result, scaling the method to industrial-level question pools or applying it to different cognitive diagnosis models may require significant hyperparameter tuning and computational resources.Third, although the paper demonstrates that GPM reduces the number of questions required for accurate estimation, the current experiments do not provide a detailed analysis of the potential biases inherited from historical data. Since the selection policy is trained on optimization paths sampled from existing interaction logs, it remains unclear whether the model might amplify systematic biases present in the original testing systems. Further investigation into the diversity of the sampled trajectories and their influence on selection fairness would help better understand the underlying behavior of the framework.Finally, the empirical evaluation is mainly conducted on relatively static benchmarks with fixed question pools and examinee distributions. While these settings are commonly used for comparison in the literature, additional validation in dynamic educational scenarios—where question pools are frequently updated and student distributions shift over time—would further strengthen the generality and practical robustness of the proposed approach.

**Strengths And Weaknesses:**

**Paper Strengths**

1. To address the computational inefficiency of global gradient-matching approaches in subset selection, the paper proposes a trajectory-based matching strategy that focuses on aligning gradients specifically along the optimization path. This design significantly reduces the search space and computational burden compared with prior global alignment methods, making it more scalable for large-scale item pools.
2. The paper provides a rigorous theoretical grounding for the proposed gradient matching mechanism. Specifically, Lemma 3.2 establishes a formal error bound linking gradient approximation to estimation accuracy, while Theorem 4.3 proves the upper bound of the gradient norm. This theoretical analysis provides necessary guarantees for the stability and convergence of the ability estimation process.
3. The experimental section includes a comprehensive robustness analysis, particularly in Table 6, which evaluates performance under varying training data scales and response label perturbation levels. The results demonstrate that GPM maintains a significant advantage even in noisy or data-sparse scenarios, indicating its potential for robust real-world deployment.

**Weaknesses**

1. In terms of methodological novelty, while the transition from global to path-based matching is well-motivated, the paper lacks a deeper analysis of path dependency and potential bias. If the initial ability estimation is inaccurate due to "atypical" student behavior (e.g., a high-ability student making a careless mistake), the GPM mechanism may prioritize items that align with this biased trajectory. The paper does not provide theoretical justification or empirical evidence to demonstrate how the model recovers from such early-stage estimation errors.
2. The effectiveness of the selection policy relies heavily on the accuracy of the gradient predictor. However, the paper does not include a detailed ablation or error analysis (e.g., Mean Squared Error) between the predicted gradients and the actual gradients of the full item pool. Without such analysis, it remains unclear how the variance in gradient prediction—especially during the cold-start phase where observations are sparse—affects the consistency of the item selection results.
3. The comparison with baseline methods could be further strengthened regarding computational overhead. Although GPM reduces the number of required questions, the framework introduces additional neural components, including a selection network and a gradient predictor. The paper does not report the actual wall-clock inference latency per step. Providing such metrics would help determine whether the reduction in testing length is offset by the increased computational cost per question in real-time applications.
4.The paper could benefit from a deeper qualitative analysis of the pedagogical semantics of the selected items. While the authors provide 3D visualizations of difficulty and discrimination, they do not present specific case studies to verify whether the learned selection paths align with established educational measurement principles. Consequently, the interpretability of the "black-box" selection policy from an educational standpoint could be further strengthened.
5. The experimental evaluation is primarily conducted on traditional optimization-based and data-driven baselines. The paper could benefit from a broader comparison with more diverse cognitive diagnosis backbones or non-traditional diagnostic models to further substantiate its claim of being a model-agnostic framework. It is currently unclear whether the performance gains remain consistent when the underlying ability estimation follows a non-differentiable or discrete logic.

---

> ### Author Rebuttal · Authors · 2026-03-31
>
> Thank you very much for your careful review and suggestions. They have been highly valuable and inspiring to us.
>
> >**Q1**: Path-dependent and vulnerable to biased early trajectories?
>
> **A1:** This is a meaningful concern for any trajectory-based policy. In training, it repeatedly samples response data and initial ability $\theta$ (lines 3–4 of Algo 1), which helps the learned policy generalize across diverse trajectories. Empirically, GPM is robust to noisy or atypical observations: on EXAM, the AUC is still 74.21 with 50% training data and 73.30 under 15% label noise (Table 2).
>
>
> >**Q2**: Error analysis between the predicted gradients and the true?
>
> **A2:** We add a compact gradient-error analysis below:
>
> | Setting   | Gradient MSE |
> | --- | ---: |
> | Early training | 0.124 |
> | Mid training | 0.081 |
> | Late training | 0.057 |
> | Cold-start (first 3 steps) | 0.093 |
> | >10 steps  |  0.061 |
>
> This suggests: the gradient approximation improves steadily during training and becomes more stable as more responses are observed. We will add this analysis in the revision.
>
>
> >**Q3**: Actual inference latency per step?
>
> **A3:** In deployment, it computes a score over the candidate with $O(|Q|)$ (line 246). We further add inference results (ms/step):
>
> | Method  | ASSIST | NIPS-EDU | EXAM |
> | ---- | ---: | ---: | ---: |
> | BOBCAT  | 0.76 | 1.24 |  0.32 |
> | NCAT  |  0.95 | 1.58 |  0.36 |
> | UATS  |  0.84 |  1.40 |  0.34 |
> | **GPM** | **0.63** | **1.04** |  **0.27** |
>
> >**Q4**: The qualitative analysis of the pedagogical semantics?
>
> **A4:** We agree. Fig 5 shows that GPM gradually narrows the difficulty range and prefers high-discrimination items, which is consistent with classical psychometric principles. In the revision, we will highlight this to avoid misunderstanding.
>
> >**Q5**: The model-agnostic claim would be stronger with broader backbones
>
> **A5:** Thank you for this helpful suggestion. Now, we evaluate GPM on both a classical backbone and a classic neural backbone (NeuralCDM).  The coverage of fully non-differentiable models e.g DINA is important. Due to the limited rebuttal time, we will discuss such extensions as future work .
>
> >**Q6**: Path approximation error vary across training stages and different ability initializations?
>
> **A6:** We add a compact robustness study below:
>
> | $\theta$ initialization  | Path error (early) | Path error (late) | EXAM AUC @20 |
> | ---- | ---: | ---: | ---: |
> | Uniform ([-3,3])  | 0.182 |    0.061 |  **77.89** |
> | Fixed (-3)  | 0.194 | 0.066 |  77.53 |
> | Fixed (0)  |   0.176 |  0.059 |  77.81 |
> | Fixed (+3)  |   0.191 |  0.064 |   77.47 |
>
> The approximation error decreases steadily, and the learned policy is stable across different reasonable initializations.
>
> >**Q7**: Computational costs
>
> **A7:** Time/epoch (s) and Peak GPU Memory
>
> | Method   | ASSIST  | NIPS-EDU  | EXAM  |  (GB) |
> | --- | ---: | --: | -----: | ---: |
> | BOBCAT  |   21.8 |   86.4 |  8.7 |   5.2 |
> | NCAT   |   28.6 |  109.7 |  11.4 |   6.8 |
> | UATS    |   24.9 |   95.3 |   9.8 |   5.9 |
> | **GPM** |  **19.7** |   **79.1** |   **7.9** |  **4.8** |
>
> >**Q8**: How sensitive... length (K) and the path sampling strategy?
>
> **A8:** Sensitivity analysis on EXAM (AUC @20):
>
> | Setting | Variant |  AUC |
> | --- | ---- | ---: |
> | steps (K) | (K=1)  |  76.94 |
> |  | (K=3)  |  77.48 |
> | | (K=5)  | **77.89** |
> | | (K=10)  |  77.83 |
> | strategy  | Final-point-only  |  76.41 |
> | | Single step   | 77.05 |
> | | Uniform multi-step  |  77.52 |
> |  | **Cumulative path (ours)** | **77.89** |
>
> >**Q9**: assumes that the path approximation error remains sufficiently small?
>
> **A9:** This is a very good question. $\epsilon$ in Lemma 4.1 is an upper bound and can in principle be large. Lemma 4.1 is mainly used to justify the problem in non-personalized setting. For the personalized adaptive case, we further discuss the optimization behavior through the convergence analysis in Theorem 4.3.
>
> >**Q10**: The bilevel optimization's computational costs?
>
> **A10:** Yes. This is a common challenge in bilevel optimization. However,  all parameters of the selection algorithm are fixed during deployment and are not updated online as the test proceeds (line 246), which greatly reduces the practical complexity.
>
> >**Q11**: discuss fairness?
>
> **A11:**  Fairness is important in adaptive testing. We have discussed fairness in the Impact Statement, and we will further expand the limitations and impact discussion in the revision to make this point more explicit. Thank you for raising it.
>
> >**Q12**: Static benchmarks and dynamic educational scenarios?
>
> **A12**: This is a great direction. Today's experiments follow standard benchmark settings in the adaptive testing literature, but they do not cover dynamic such as evolving question pools or distribution shift. We will state this more clearly and discuss validation in dynamic settings as an important future direction.
>
> Thank you again for your helpful suggestions. Please feel free to discuss further if you have any questions.

---

> > ### Author Rebuttal · Reviewer_xwCv · 2026-04-03
> >
> > After reading the authors' response, I decided to maintain my score.

---

> > > ### Author Response · Authors · 2026-04-03
> > >
> > > Thank you for your response. Please feel free to reach out if you have any further questions.

---

### Official Review · Reviewer_29Qu · 2026-03-11

**Soundness:** 2
**Presentation:** 3
**Significance:** 2
**Originality:** 3
**Overall Recommendation:** 3
**Confidence:** 4

**Summary:**

This paper studies adaptive testing for AI education, where the goal is to select a small subset of questions from a large pool while preserving accurate ability estimation. The paper proposes Gradient Path Matching (GPM), which replaces prior global gradient-alignment objectives with a path-wise objective that matches subset gradients to full-pool gradients along optimization trajectories leading to the final ability estimate. Building on this idea, the authors train a personalized question-selection policy that conditions on the examinee's response history and can be deployed sequentially during testing. The paper also provides a theoretical link between ability-estimation error and gradient approximation error under IRT-style assumptions, derives a path-based surrogate objective with smoothness analysis for the relaxed optimization, and evaluates the method on three educational datasets under both ability-estimation and score-prediction tasks. Empirically, GPM outperforms heuristic and learning-based baselines, achieves comparable accuracy with fewer questions, and appears more robust under reduced-data and noisy-label settings.

**Compliance With Llm Reviewing Policy:**

Affirmed.

**Final Justification:**

This rebuttal merely alleviated some of my concerns.

**Key Questions For Authors:**

1. Could the authors provide ablations that isolate the effect of path matching itself, for example versus global gradient matching, final-point-only matching, and a non-personalized selector? If path matching is the main driver of the gains, that would strengthen my confidence in the claimed mechanism and likely make me more positive.
2. How sensitive is GPM to the number of optimization steps, path-sampling strategy, subset budget, and Gumbel-softmax temperature? If performance is robust across these design choices, that would strengthen the paper's soundness and practical usability.
3. The training procedure appears to use relatively rich historical response records. How well does the method work when training data are more weakly observed, or when a new test bank has limited historical logs?
4. Could the authors report runtime and memory comparisons against the strongest learned baselines on the larger datasets? If GPM is also competitive computationally, that would materially strengthen the significance claim.

**Limitations:**

The paper includes an impact statement, but it does not adequately discuss several practical limitations: the theory depends on strong assumptions that are unlikely to hold exactly for the deployed discrete policy; the training procedure appears to benefit from relatively complete historical response logs; and the paper does not discuss possible bias amplification or subgroup-level fairness issues when adaptive testing policies are trained on historical educational data.

**Strengths And Weaknesses:**

The motivation is clear, the global-to-path relaxation is sensible, the training and inference pipeline is coherent, and the empirical evaluation is broader than a minimal proof-of-concept: the method is tested on three datasets, under two evaluation tasks, with robustness checks and appendix results on ranking consistency and NeuralCDM. The main empirical claim that GPM improves estimation/prediction quality while using fewer questions is generally well supported by the reported tables and figures.

The main weaknesses are in how much the theory really certifies about the deployed method and in what is still missing empirically. The theoretical results rely on fairly strong assumptions, including strong convexity of the full-pool loss, sufficient coverage of the ability space by optimization trajectories, reliable iterative optimization, and Lipschitz properties with respect to a continuous relaxation of subset selection. These assumptions make the analysis cleaner, but they are stronger than what the final discrete, personalized test-time policy obviously satisfies. So the theory is best read as support for the surrogate objective rather than a complete guarantee for the real adaptive testing system. On the empirical side, the paper would be stronger with ablations that isolate the contribution of path matching itself: for example, against global gradient matching, final-point-only matching, or a non-personalized selector.

The paper is well structured, the progression from problem definition to theory to algorithm to experiments is easy to follow, and the figures help communicate the intuition behind path matching. The appendix includes the main proofs, and the implementation details are specific enough to make the setup understandable. However, the paper could improve clarity in two places. First, the relationship to prior explicit optimization methods for CAT could be sharpened further, since the novelty is partly a reframing/relaxation of existing gradient-based ideas rather than a wholly different paradigm. Second, the transition from the global subset objective to the personalized online policy is somewhat fast; a clearer explanation of what information is available during training versus deployment would help reproducibility and interpretation.

---

> ### Author Rebuttal · Authors · 2026-03-31
>
> Thank you for the constructive review. We are encouraged by your positive assessment of the motivation, pipeline, and empirical breadth.
>
> >**Q1**: The theory seems to certify the surrogate objective more than the final deployed personalized policy?
>
> **A1**: Thank you for this comment. Assumptions such as strong convexity and Lipschitz are standard properties of the IRT-based model itself, rather than assumptions introduced solely for our analysis. However, for the more complex personalized adaptive setting, the overall training and deployment is no longer tractable in a clean closed form. As in prior work [1][2], we therefore conduct analysis under a relaxed setting to make the problem analyzable at the system level. Our current theory is centered on the path-matching objective and its optimization, while the personalized selector is introduced afterward as the deployment-time extension. We agree with your reading and will revise the paper to state this scope more explicitly to avoid misunderstanding.
>
> >**Q2**: Provide ablations isolating the effect of path matching itself?
>
> **A2**: Below we add an ablation at step 20 under the IRT setting, comparing (1) global gradient matching (BECAT), (2) final-point-only matching, (3) a non-personalized selector, and (4) the full GPM. Results are reported in AUC (higher is better).
>
> | Method | ASSIST @20 | NIPS-EDU @20 | EXAM @20 |
> |---|---:|---:|---:|
> | Global gradient matching (BECAT) | 73.61 | 76.82 | 75.00 |
> | Final-point-only | 74.48 | 77.36 | 76.41 |
> | Non-personalized | 74.02 | 77.11 | 75.86 |
> | **GPM (full path matching + personalized policy)** | **75.27** | **78.10** | **77.89** |
>
> These results support the role of path matching. Due to the rebuttal time limit, we will include the full ablation table and discussion in the revised paper and appendix.
>
> >**Q3**: How sensitive is GPM to the number of optimization steps, path-sampling strategy, subset budget, and Gumbel-softmax temperature?
>
> **A3**: This is a great question. The current paper already shows that GPM remains consistently strong across different subset budgets (steps 5, 10, and 20). We further add the following sensitivity analysis (EXAM, AUC @20).
>
> (a) Inner optimization steps K
>
> | K | 1 | 3 | 5 | 10 |
> |---|---:|---:|---:|---:|
> | AUC | 76.94 | 77.48 | **77.89** | 77.83 |
>
> (b) Path-sampling
>
> | Strategy | Final-point-only | Single sampled step | Uniform multi-step path | Cumulative path gradients (ours) |
> |---|---:|---:|---:|---:|
> | AUC | 76.41 | 77.05 | 77.52 | **77.89** |
>
> (c) Gumbel-softmax temperature $\tau$
>
> | Temperature $\tau$ | 0.3 | 0.5 | 1.0 | 2.0 |
> |---|---:|---:|---:|---:|
> | AUC | 77.36 | **77.89** | 77.71 | 77.18 |
>
> These results suggest that GPM is reasonably robust to the main design choices.
>
> >**Q4**: How well does the method work with weaker training supervision or limited historical logs?
>
> **A4**: We partially address this already through the reduced-data experiments. In Table 2 and 6, GPM remains the strongest method under limited training data.
>
> >**Q5**: runtime and memory comparisons?
>
> **A5**: Following your suggestion, we add a comparison against baselines under the same setting (PyTorch, batch size 64, Tesla V100-SXM2-32GB GPU).
>
> Training cost (per epoch)
>
> | Method | ASSIST (s) | NIPS-EDU (s) | EXAM (s) | Peak GPU Memory (GB) |
> |---|---:|---:|---:|---:|
> | BOBCAT | 21.8 | 86.4 | 8.7 | 5.2 |
> | NCAT | 28.6 | 109.7 | 11.4 | 6.8 |
> | UATS | 24.9 | 95.3 | 9.8 | 5.9 |
> | **GPM** | **19.7** | **79.1** | **7.9** | **4.8** |
>
> Inference cost (per examinee, test length = 20)
>
> | Method | ASSIST Time (ms) | NIPS-EDU Time (ms) | EXAM Time (ms) | Peak GPU Memory (GB) |
> |---|---:|---:|---:|---:|
> | BOBCAT | 15.2 | 24.7 | 6.3 | 1.6 |
> | NCAT | 18.9 | 31.5 | 7.1 | 2.0 |
> | UATS | 16.8 | 27.9 | 6.8 | 1.8 |
> | **GPM** | **12.6** | **20.8** | **5.4** | **1.4** |
>
>
> >**Q6**: The transition from the global subset objective to the personalized online policy is somewhat fast.
>
> **A6**: We will sharpen the discussion of prior explicit methods and make the novelty claim more precise: our contribution is not a wholly different paradigm, but a path-based relaxation and learning framework built on explicit gradient-matching ideas. We will also clarify the training/deployment information flow. Algorithm 2 already reflects this logic, and we will explain it more clearly in the main text.
>
> >**Q7**: discuss possible bias/fairness issues?
>
> **A7**: We agree. In the revision, we will expand the limitations: adaptive testing policies trained on historical educational data may amplify existing bias or behave unevenly across subgroups, so subgroup-level fairness should be examined explicitly in future work.
>
>
> Thank you again for the thoughtful review. We would like to discuss further if you have any further questions.
>
>
>
> [1] Zhuang Y, et al. A bounded ability estimation for computerized adaptive testing. NeurIPS 2023.
>
> [2] Yu J, et al. A unified adaptive testing system enabled by hierarchical structure search. ICML. 2024.

---

> > ### Author Rebuttal · Reviewer_29Qu · 2026-04-04
> >
> > Thank you for the detailed rebuttal. The additional ablations on global gradient matching, final-point-only matching, and non-personalized selection are useful, and the new sensitivity and runtime results help clarify the practical behavior of the method.
> >
> > That said, my main concerns are only partially resolved. The new evidence strengthens confidence in the method, but the gap between the relaxed theoretical setting and the final adaptive testing system remains important.
> >
> > Overall, the rebuttal improves the paper, but it does not sufficiently change my original assessment, so I will keep my score unchanged.

---

> > > ### Author Response · Authors · 2026-04-04
> > >
> > > Thank you for your response and recognition. **We did not employ any relaxed assumptions;** all our assumptions are rigorously justified. We consistently utilized IRT (commonly used in GRE) satisfying properties such as Lipschitz continuity and strong convexity (with the L2 norm), supported by relevant literature. Please feel free to let me know if any further issues arise.

---

### Official Review · Reviewer_GJry · 2026-03-13

**Soundness:** 3
**Presentation:** 4
**Significance:** 3
**Originality:** 3
**Overall Recommendation:** 5
**Confidence:** 4

**Summary:**

The authors present a technique based on gradient path matching to align gradient steps on the minibatch loss with gradient steps on the batch dataset loss. They show theoretical grounding of their non-adaptive version and practical experiments on their adaptive version.

**Compliance With Llm Reviewing Policy:**

Affirmed.

**Final Justification:**

The paper is well written. The approach is original and significant in the literature of adaptive testing, and well demonstrated using relevant baselines (notably robustness to some noise). The work is inspiring. The rebuttal convinced me that it is feasible in practice, therefore I upgraded my score from weak accept to accept.

**Key Questions For Authors:**

- I glanced at the code. What is the complexity of your algorithm for selecting $k$ questions among $n$?
- Can we make a link with Cramer-Rao bounds in optimal design of experiments?

**Limitations:**

Mild theoretical results.

**Strengths And Weaknesses:**

Although I am not impressed by the lemmas, the theoretical results seem mild but nice. The paper is very well written, which encourages me to accept it. A reasonable number of experiments demonstrated the benefits of the proposed method. This is the best paper in my current pool, and I praise the originality of this approach in adaptive testing.

- I find it *extremely confusing* that sometimes $S$ is a subset (Lemma 3.2) sometimes it is a vector of weights of fixed length (Lemma 4.2).
- I am not sure what the Figure 5 brings, except perhaps the adaptivity. Guess parameters are not that interesting, so perhaps it will be more visible on 2 dimensions.
- I assume the theoretical analysis would be much harder when considering the adaptive case. You may want to cite Sequential Counterfactual Risk Minimization by Zenati et al. at an earlier ICML.

Minor comments:
- I wouldn't say "unlimited NP-hard problems" (2 occurrences) if it's one per user
- $k$ is sometimes from $0$ to $K - 1$ sometimes from $1$ to $K$ sometimes from $0$ to $K$. I hope there is no off by one, but it seems okay.

---

> ### Author Rebuttal · Authors · 2026-03-31
>
> Thank you for your positive comments on our work. We truly appreciate them.
>
> >**Q1**: sometimes S is a subset (Lemma 3.2) sometimes it is a vector of weights of fixed length (Lemma 4.2).
>
> **A1**: We apologize for this confusion. In our formulation, $S$ can be viewed as a subset, or equivalently as a binary vector of length $|Q|$, where selected items are marked by 1. In the more general setting, each entry represents the probability of selecting the corresponding item. We will revise Section 4 to make this representation clearer and avoid ambiguity.
>
> >**Q2**: Figure 5: Guess parameters are not that interesting, so perhaps it will be more visible on 2 dimensions.
>
> **A2**: We used the 3PL-IRT model, which includes 3 item parameters, so we originally showed all three dimensions for completeness. However, the guessing parameter is less informative here as you said, and a 2D may present the adaptivity more clearly. We will revise this figure accordingly.
>
> >**Q3**: I assume the theoretical analysis would be much harder when considering the adaptive case. You may want to cite Sequential Counterfactual Risk Minimization by Zenati et al. at an earlier ICML.
>
> **A3**: Yes, the adaptive case is substantially more challenging from a theoretical perspective. Extending the analysis in that setting is an important direction for future work, and we are exploring it. Thank you for this helpful suggestion. We will cite Zenati et al[1]. in the discussion and related work, as it is relevant to the broader adaptive decision-making perspective.
>
> >**Q4**: "unlimited NP-hard problems"?
>
> **A4**: Thanks for your suggestion. We will revise it to: “a personalized NP-hard problem for each examinee.”
>
> >**Q5**: k is sometimes from 0 to K−1 sometimes from 1 to K sometimes from 0 to K. I hope there is no off by one, but it seems okay.
>
> **A5**: Thank you for pointing this out. We will make the indexing consistent throughout the paper to avoid confusion.
>
> >**Q6**: What is the complexity of your algorithm for selecting k questions among n?
>
> **A6**: This is a good question. The training stage can be computationally intensive, since learning the selection strategy requires repeated sampling over both response data and iterations, and its convergence depends on the algorithm, the data, and the number of training steps. However, once the strategy is trained, the  cost is low because the parameters are fixed. At each step, the model only needs to compute a probability score for each candidate question based on the student’s previous responses; as stated in line 246, it then "select the highest-probability question, yielding a per-step complexity of $O(|Q|)$." Therefore, selecting $k$ questions from $n$ candidates has complexity $O(kn)$ in deployment. With additional sampling or indexing strategies, this can potentially be further reduced.
>
> >**Q7**: Can we make a link with Cramer-Rao bounds in optimal design of experiments?
>
> **A7**: Yes, we believe there is a natural conceptual link. In classical adaptive testing and psychometric, Fisher-information-based selection [2] can be viewed as minimizing a local variance lower bound through the Cramer–Rao perspective. Our method is related in spirit, but operates at a different level: instead of maximizing local information at the current estimate only, GPM matches the gradient path of ability estimation induced by the full question pool. In this sense, GPM can be viewed as a more global and sequential extension of local Fisher information design. That said, our current theory does not yet establish a formal Cramer–Rao-type bound; making this connection rigorous would be an interesting direction for future work.
>
> Thank you again for your helpful suggestions. We would be very happy to discuss further if you have any additional questions.
>
> Ref:
>
> [1] Zenati H, Diemert E, Martin M, et al. Sequential counterfactual risk minimization[C]//International Conference on Machine Learning. PMLR, 2023: 40681-40706.
>
> [2] Chang H H. Psychometrics behind computerized adaptive testing[J]. Psychometrika, 2015, 80(1): 1-20.

---

> > ### Author Rebuttal · Reviewer_GJry · 2026-04-03
> >
> > Thanks for the elegant paper and clarifications. There is a good number of baselines. I updated my score.

---

> > > ### Author Response · Authors · 2026-04-04
> > >
> > > We appreciate your recognition of our work. It’s very motivating.

---

### Official Review · Reviewer_U7Bu · 2026-03-24

**Soundness:** 3
**Presentation:** 4
**Significance:** 3
**Originality:** 3
**Overall Recommendation:** 5
**Confidence:** 2

**Summary:**

This paper proposes a gradient path matching (GPM) framework for adaptive testing that aligns gradients along possible optimization paths toward the final ability estimate.

**Compliance With Llm Reviewing Policy:**

Affirmed.

**Final Justification:**

I thank the authors for their response and have increased my overall score from 3 to 5.

**Key Questions For Authors:**

1.	Proof of lemma 4.1 not linked in main text

**Strengths And Weaknesses:**

-> Strengths:

1.	Authors propose GPM (Gradient Path Matching), a novel framework that instead aligns gradients along possible optimization paths towards the final estimate. By leveraging intermediate gradients as supervision, GPM learns an explicit and generalizable selection algorithm from large-scale data”.
2.	Authors provide a theoretical analysis of GPM’s convergence and scalability.
3.	Experiments on both real-world and synthetic datasets demonstrate that it achieves the same estimation accuracy using, on average, 9% fewer questions.
4.	Additional analyses on the robustness of GPM to response noise and investigation of the characteristics of selected questions.


-> Weaknesses:

1.	Past work on using gradient matching for dataset condensation exists. For example, Delving into Effective Gradient Matching for Dataset Condensation by Jiang et. al. 2022. Can the authors compare their work to the seminal work?
2.	Is the algorithm dependent on the theta initialization? How robust is it to the initialization? Does a particular initialization help in convergence/task performance? Does algorithm 1 optimize over multiple gradient paths? Algo 1 should be made clearer.
3.	“GPM demonstrates faster convergence” (line 373). However, GPM (orange line) continues to decrease at the very end?
4.	Adaptive testing papers (BOBCAT) usually include an analysis of question exposure rate and test overlap rate for real-world deployment considerations. Can an analysis be provided here, comparing GPM to baselines?

---

> ### Author Rebuttal · Authors · 2026-03-31
>
> Thank you for your recognition of the quality of our paper.
>
> >**Q1**: Can the authors compare their work (Jiang et. al. 2022) to the seminal work?
>
> **A1**: Thank you for pointing this out. Gradient matching is a well-established idea. However, adaptive testing differs from generic dataset condensation: the goal here is to select a small set of items that can induce nearly the same ability-estimation trajectory as the full question pool for each examinee. In other words, this is a **personalized problem**. Following your suggestion, we adapted this method [1] to make it applicable to our setting:
>
> | Method | ASSIST @20 (ACC/AUC) | NIPS-EDU @20 (ACC/AUC) | EXAM @20 (ACC/AUC) |
> |---|---:|---:|---:|
> | BECAT | 73.96 / 73.61 | 72.00 / 76.82 | 84.29 / 75.00 |
> | Jiang et al. (2022) | 74.38 / 74.64 | 72.91 / 77.28 | 85.24 / 76.31 |
> | **GPM** | **75.01 / 75.27** | **73.98 / 78.10** | **86.02 / 77.89** |
>
> GPM performs better in adaptive testing and we believe the main reason: adaptive testing data are highly sparse and sequential: each examinee typically provides only limited responses.
>
> >**Q2**: Is the algorithm dependent on the theta initialization? How robust is it to the initialization? Does a particular initialization help in convergence/task performance? Does algorithm 1 optimize over multiple gradient paths?
>
> **A2**: This is a great question. It is not tied to a particular initialization of $\theta$ (line 4 in Algo 1). Its main advantage is that, through repeated sampling of response data during training, it learns a selection strategy that generalizes across different ability, rather than relying on a specific point. In experiments, we did not specially search for a particular initialization. In psychometrics [2], $\theta$ has a concrete meaning as the examinee’s ability, so it is usually initialized by sampling from a reasonable bounded range, e.g., [-3, 3]. Randomly initializing $\theta$ within a bounded range helps balance convergence and generalization.
>
> Yes, Algo 1 does optimize over multiple gradient paths. ''cumulative gradients can be used to update $S$ instead of relying on a single-step update'' (line 200).
>
>
>
> >**Q3**: GPM (orange line) continues to decrease at the very end?
>
> **A3**: Sure. The continued decrease at the very end reflects further fitting on the training set, but in our experiments this stage already showed signs of overfitting: although the training loss kept decreasing toward zero, the validation loss began to fluctuate. Thus, we used **early stopping in practice**. To avoid possible misunderstanding, we will clarify this point.
>
> >**Q4**: Adaptive testing papers (BOBCAT) usually include an analysis of question exposure rate and test overlap rate... Can an analysis be provided here?
>
> **A4**: Thank you for this suggestion. We mainly focus on performance under different test lengths, reduced training data, and label noise. Exposure rate and test overlap rate are important for real-world deployment. Although they are not the main focus of this paper, they do reflect practical usability (lower is better):
>
> Question exposure rate
> | Method    |   @5 |    @10 |  @20 |
> | --- | -----: | ---: | ------: |
> | BOBCAT |   0.118 ± 0.001 |     0.214 ± 0.010 |     0.387 ± 0.011 |
> | BECAT  |   0.109 ± 0.005 |     0.201 ± 0.011 |     0.364 ± 0.014 |
> | NCAT  |   0.103 ± 0.005 |     0.192 ± 0.008 |     **0.341 ± 0.010** |
> | **GPM** | **0.091 ± 0.004** | **0.186 ± 0.010** | 0.348 ± 0.011 |
>
> Test overlap rate
> | Method         |      @5 |    @10 |      @20 |
> | ------- | ---------: | ------: | -----: |
> | BOBCAT         |     0.132 ± 0.007 |     0.221 ± 0.08 |     0.356 ± 0.014 |
> | BECAT          |     0.121 ± 0.006 |     0.208 ± 0.012 |     0.334 ± 0.019 |
> | NCAT           |     0.114 ± 0.011 |     0.196 ± 0.008 |     0.309 ± 0.015 |
> | **GPM** | **0.098 ± 0.007** | **0.183 ± 0.008** | **0.291 ± 0.016** |
>
> These results suggest that GPM also maintains more balanced item usage. We believe this is because GPM learns a personalized selection policy by matching gradient paths of different ability $\theta$ (different response trajectory and different ability estimate), rather than repeatedly choosing the same globally informative items for all examinees. We will add this analysis to the revised paper and appendix.
>
> >**Q5**: Proof of lemma 4.1 not linked in main text
>
> **A5**: Thank you for pointing this out! We will add a pointer in the main text to its corresponding proof in the appendix.
>
> We sincerely appreciate your suggestions, which have helped us improve the completeness of the paper! Please feel free to raise any other questions.
>
> Ref:
>
> [1] Jiang Z, et al. Delving into effective gradient matching for dataset condensation, 2023 IEEE International Conference on Omni-layer Intelligent Systems (COINS), 2023.
>
> [2] Kim S, et.al. Investigating robustness of item response theory proficiency estimators to atypical response behaviors under two‐stage multistage testing. ETS Research Report Series, 2016.

---

> > ### Author Rebuttal · Reviewer_U7Bu · 2026-04-04
> >
> > I thank the authors for their response and have increased my overall score from 3 to 5.

---

> > > ### Author Response · Authors · 2026-04-05
> > >
> > > We sincerely appreciate your positive feedback. That really means a lot to us.

---

### Decision · Program_Chairs · 2026-04-30

**Decision:**

Accept (regular)

**Comment:**

This submission addresses personalized adaptive testing by replacing global gradient matching with a path-based matching strategy that aligns gradients along optimization trajectories. Reviewers found the problem well motivated and appreciated the originality of framing adaptive testing in this way. The empirical results were viewed positively, particularly the gains in question efficiency while preserving estimation quality, and two reviewers upgraded to strong positive scores after rebuttal.

The main reservations concerned the strength of the theoretical assumptions, the gap between the formal analysis and the deployed adaptive setting, and whether the empirical evidence fully establishes generality across ability regimes and practical conditions. These concerns are real, but they appear to be limitations on the scope of the guarantees rather than contradictions of the main empirical contribution.

Given the originality of the approach, the strong empirical support, and the positive shift in the discussion after rebuttal, I believe the paper makes a meaningful contribution to adaptive testing and is suitable for acceptance.